

# Systematic review of predictive maintenance and digital twin technologies challenges, opportunities, and best practices

Nur Haninie Abd Wahab[1,2], Khairunnisa Hasikin[1,3], Khin Wee Lai[1], Kaijian Xia[1,4], Lulu Bei[5], Kai Huang[6] and Xiang Wu[1,7]

[1] Department of Biomedical Engineering, Faculty of Engineering, Universiti Malaya, Kuala Lumpur, Malaysia
[2] Engineering Services Division, Ministry of Health Malaysia, Putrajaya, Malaysia
[3] Center of Intelligent Systems for Emerging Technology, Faculty of Engineering, Universiti Malaya, Kuala Lumpur, Malaysia
[4] Affiliated Changshu Hospital, Soochow University Changshu, Jiangsu, China
[5] School of Information Engineering, Xuzhou University of Technology, Xuzhou, China
[6] JiangSu XCMG HanYun Technologies Co., LTD., Xuzhou, China
[7] School of Medical Information & Engineering, Xuzhou Medical University, Xuzhou, China

Corresponding authors
Khairunnisa Hasikin,
khairunnisa@um.edu.my
Lulu Bei, beilulu66@126.com

## ABSTRACT

**Background:** Maintaining machines effectively continues to be a challenge for industrial organisations, which frequently employ reactive or premeditated methods. Recent research has begun to shift its attention towards the application of Predictive Maintenance (PdM) and Digital Twins (DT) principles in order to improve maintenance processes. PdM technologies have the capacity to significantly improve profitability, safety, and sustainability in various industries. Significantly, precise equipment estimation, enabled by robust supervised learning techniques, is critical to the efficacy of PdM in conjunction with DT development. This study underscores the application of PdM and DT, exploring its transformative potential across domains demanding real-time monitoring. Specifically, it delves into emerging fields in healthcare, utilities (smart water management), and agriculture (smart farm), aligning with the latest research frontiers in these areas.
**Methodology:** Employing the Preferred Reporting Items for Systematic Review and Meta-Analyses (PRISMA) criteria, this study highlights diverse modeling techniques shaping asset lifetime evaluation within the PdM context from 34 scholarly articles.
**Results:** The study revealed four important findings: various PdM and DT modelling techniques, their diverse approaches, predictive outcomes, and implementation of maintenance management. These findings align with the ongoing exploration of emerging applications in healthcare, utilities (smart water management), and agriculture (smart farm). In addition, it sheds light on the critical functions of PdM and DT, emphasising their extraordinary ability to drive revolutionary change in dynamic industrial challenges. The results highlight these methodologies' flexibility and application across many industries, providing vital insights into their potential to revolutionise asset management and maintenance practice for real-time monitoring.
**Conclusions:** Therefore, this systematic review provides a current and essential resource for academics, practitioners, and policymakers to refine PdM strategies and expand the applicability of DT in diverse industrial sectors.

## INTRODUCTION

The dependency of modern industry on machine tools that are becoming increasingly complicated presents a challenge for those in charge of maintenance planning (*Aivaliotis, Georgoulias & Chryssolouris, 2019*; *Zhang et al., 2022*). Initial research conducted by *Xiong et al. (2021)* and *You et al. (2022)* discovered that equipment failures during operation may be unpredictable and may result in a decrease in production efficiency. Under the direst of circumstances, the machinery will be rendered inoperable, resulting in the destruction of machinery and the loss of human life (*van Dinter, Tekinerdogan & Catal, 2022*; *Zhai & Qiao, 2020*). If the rapid intervention is not performed, the machine may encounter unforeseen failures that result in low precision, production downtime, or even catastrophic loss (*Centomo, Dall'Ora & Fummi, 2020*; *Luo et al., 2020*; *Mubarak et al., 2022*). In the event that the concerns are not promptly remedied, there is an increased likelihood of the occurrence of catastrophic events.

Over the years, modern machine tools have become increasingly complex, making it difficult to anticipate when failures may occur. Using advanced analytics and sensor data, PdM is implemented to monitor the machinery's health in real-time. This enables maintenance teams to detect potential problems early and resolve them prior to their escalation into catastrophic failures. As opposed to responding to problems reactively or doing significant periodic preventive maintenance, current PdM trends encourage proactive maintenance (*Hosamo et al., 2022*; *Nunes et al., 2023*; *Ong et al., 2022*; *Werner, Zimmermann & Lentes, 2019*; *Xiong et al., 2021*). The equipment maintenance pattern has evolved due to the optimised selection of passive generation to active methods. Simultaneously, it exhibits superior performance compared to traditional maintenance approaches. Maintenance tasks that offer no value are diminished. The prolonged operational lifetime and enhanced production efficiency may be attributed to the heightened reliability and stability of the equipment.

This data-driven approach has enabled the foundation of continuous improvement through data. In addition, the Industry 4.0 revolution trends aspire to widespread adoption of DT. A wide range of industrial sectors have created virtual representations of physical machines or systems using DT in various applications such as design, production, manufacturing, and maintenance. Among these, maintenance is one of the applications that has attracted the most attention from researchers (*Errandonea, Beltrán & Arrizabalaga, 2020*). The term "DT" refers to two-way communication between a virtual platform and a physical asset. This makes it possible to ensure operations while also monitoring performance in real-time (*Altun & Tavli, 2019*; *Cabeza-Gil et al., 2023*; *Chung & Jung, 2023*; *Haleem et al., 2023*; *van Dinter, Tekinerdogan & Catal, 2022*). The real-time capability has allowed DT to notify repairs and maintenance by providing a detailed virtual model of equipment and aiding maintenance teams in optimising maintenance schedules and procedures, thus reducing the need for unnecessary repairs and maintenance tasks.

The concept of the digital twins (DT) makes it possible to accurately assess the condition of the equipment. For this reason, the DT concept is appearing more frequently in industrial applications in connection with predictive maintenance (PdM). It is a technique for forecasting the lifespan of vital components according to inspections or diagnoses.

Due to the recent technological advancements in Artificial Intelligence (AI), including machine learning (ML) and deep learning (DL), the Internet of Things (IoT), computer vision, and DT, there are numerous prospects for PdM utilising DT (*Avornu et al., 2022*; *van Dinter, Tekinerdogan & Catal, 2022*). DT can be employed in conjunction with ML algorithms to improve PdM capabilities. These algorithms can be trained using historical data collected from sensors and other sources to identify patterns and anomalies in an asset's behaviour. Additionally, the algorithms can then be used to predict potential failures or breakdowns and provide maintenance and repair recommendations. DT can provide a virtual model of an asset, enabling simulation and testing of various maintenance scenarios and strategies. Moreover, these simulation results can be used to optimise and improve ML algorithms, making them more accurate and effective. Furthermore, ML algorithms can be used in real-time to analyse data from DT, allowing for continuous monitoring and PdM. This can aid in lowering downtime, increasing asset reliability, and optimising maintenance schedules and procedures.

This study performs a systematic literature review (SLR) on integrating PdM and DT to identify and summarise the significant study findings. The authors aim to select relevant information regarding PdM using DT and discuss the feasibility of transforming other major fields that require real-time monitoring, such as healthcare, utilities (smart water management), and agriculture (smart farm). In addition, the authors use inclusion and exclusion criteria to analyse and filter data, including using a software programme called VOSviewer to create maps from network data and visualise and examine maps. The outcome of PdM employing DT is crucial for the DT model, data integration, and implementation, as it can result in more precise model predictions. In addition, substantial research on PdM using DT for prediction based on real-time performance monitoring and equipment operation assurance is still lacking despite the growing interest and use (*Errandonea, Beltrán & Arrizabalaga, 2020*; *Falekas & Karlis, 2021*; *van Dinter, Tekinerdogan & Catal, 2022*; *You et al., 2022*; *Zhong et al., 2023*).

Significantly, this integration has not thoroughly examined key industries such as utilities (smart water management), healthcare, and agriculture (smart farm), highlighting the need for a targeted exploration of these undiscovered areas. The most evident deficiency in current research is the imbalanced focus on certain technical sectors, resulting in a lack of comprehension of the possible applications and difficulties in vital social areas. Utilities (smart water management), which are crucial for maintaining infrastructure and healthcare, are essential for public well-being and need careful consideration in integrating PdM and DT. Similarly, the agriculture (smart farm) sector, which is essential for providing nourishment, has not fully explored the potential of these technologies to optimise farming processes and machinery.

Utilities (smart water management), being essential components of infrastructure networks, need reliable and continuous functioning systems such as water supply and

**Table 1 Mapping between identified research gaps and the potential research questions.**

| Research gaps | Research questions |
|---|---|
| Gaps in application scalability and real-time analytics | **RQ1:** How can the use of predictive maintenance (PdM) and digital twins (DT) be expanded to address large-scale industrial applications while maintaining real-time performance? |
| | **RQ2:** In what ways can innovative approaches or frameworks be devised to optimise the computational efficacy of real-time DT simulations, specifically for expansive and intricate industrial settings? |
| | **RQ3:** How do machine learning (ML) and distributed processing facilitate real-time analytics PdM, particularly in industries with extensive and geographically dispersed infrastructures? |
| Strategies for seamless integration that are insufficient | **RQ4:** What are the most effective strategies for ensuring the interoperability and standardisation of DT and PdM solutions across industries and organisations? |
| | **RQ5:** How can ML algorithms be optimised to handle increasing volumes of data generated by PdM and DT systems without compromising performance? |
| | **RQ6:** What are the best practices and strategies for extending PdM and DT applications from traditional engineering sectors to critical domains such as healthcare, utilities (smart water management), and agriculture (smart farm)? |
| Inadequate integration of human-centric decision-making | **RQ7:** What collaborative frameworks and methodologies can foster effective communication and collaboration between PdM and DT systems and human decision-makers in applications such as healthcare, utilities (smart water management), and agriculture (smart farm)? |
| | **RQ8:** What strategies can be employed to bridge the gap between data-driven decision-making from PdM and DT systems and the human understanding of organisational objectives and strategies? |

electricity grids. Integrating PdM and DT may avoid downtime, maintaining the dependability and efficiency of these crucial systems. Similarly, in the healthcare field, where the utmost importance is placed on dependability for the well-being of patients, incorporating PdM and DT technologies can completely transform maintenance procedures, ensuring continuous and uninterrupted healthcare services. Meanwhile, the agriculture (smart farm) sector, which is often disregarded in terms of technical progress, offers unexplored possibilities. The agricultural industry has the potential to significantly increase production while reducing resource waste, in line with the principles of sustainable agriculture through the implementation of PdM and DT.

However, the lack of comprehensive frameworks for interoperability and standardisation across several sectors impedes the smooth integration and deployment of PdM and DT. Furthermore, the use of real-time analytics in extensive industrial environments presents notable obstacles in terms of processing efficiency and scalability, which hinder their efficient utilisation. Another factor that is often missed is the integration of human-centric PdM and DT insights into decision-making procedures. The ability to convert intricate insights into practical choices is essential for the planning and implementation of maintenance tasks. The lack of intuitive interfaces and transparent decision-making processes hinders the realisation of the potential advantages offered by these technologies.

Therefore, identifying these gaps in research raises important concerns that should be investigated, providing possibilities to modify discoveries and use insights in numerous manners. This emphasises the necessity of identifying and delineating new study inquiries that can drive subsequent inquiry and advancement in the subject. Consequently, these

issues stimulate knowledge exploration and practical application, providing a framework for future study efforts. The mapping between identified research gaps and the corresponding potential research questions is presented in Table 1.

These research questions cover various aspects of the incorporation of DT and PdM technologies and can serve as a catalyst for future research and development in this area. Identifying PdM with DT can aid in managing equipment maintenance by allowing problems to be detected before they occur and, ideally, resolved before they become severe. In addition, the authors discuss the applicability of PdM and DT in emerging disciplines as technology has evolved. To the best of the authors' knowledge, none of the included studies contributed to the topic. Hence, this study aimed to determine the significance of PdM utilising DT by systematically reviewing prior research.

# SURVEY METHODOLOGY

## Literature search

The systematic literature search was conducted using the stated standard PRISMA technique for the evaluation and rigorous analysis of articles in the database search engines (*Page et al., 2021*). Additionally, the inclusion and exclusion procedures from the pertinent recent research were thoroughly followed, as tabulated in Table 2. The analysis of included studies was identified to meet the goal of a systematic review in the field. Only the most recent research articles and conferences in the subject area were considered for inclusion to reduce the possibility of including irrelevant topics. In addition, only publications written in English were included to aid the analytic process. Review articles are excluded from this study to enhance the rigour, comprehensiveness, and dependability of the included research. This exclusion is intended to guarantee that the selected research has undergone a more stringent degree of review and adheres to conventional standards for quality.

The articles were chosen within the period of 2018 to 2023 to ensure the incorporation of the most current insights and findings within the field. Given the rapid evolution of this area, recent studies are pivotal in capturing the latest advancements. Due to the limited volume of articles available on this specific topic in the selected databases, most selected articles are in the span commencing from 2019 to November 2023. There is a deliberate exclusion of articles before 2018 due to the scarcity of literature within these databases during that time frame.

## Resources

The literature for this systematic review of PdM investigations employing DT was primarily obtained from Web of Science and Scopus. Additionally, the literature search was extended to six other prominent academic databases, namely IEEE Xplore, Science Direct, Medline Complete, Emerald, Springer Link, and Dimensions.

Figure 1 illustrates the trend of scholarly publications from 2018 to 2023, obtained from a comprehensive analysis of eight databases. This trend demonstrates a consistent rise in the use of developing technologies over consecutive years, as observed by the growing recognition among researchers. However, it is worth noting that in the year 2023, this study remains unfinished at the conclusion of the year. Hence, a decline in this data is

**Table 2 The inclusion and exclusion criteria for database searches.**

| Criterion | Eligibility | Exclusion |
|---|---|---|
| Literature type | Journal (research articles) and conference proceeding | Journal (review articles), chapter in a book, book, book series |
| Language | English | Non-English |
| Timeline | Between 2018 and 2023 | <2018 |
| Subject area | Related to PdM and DT | Other than PdM and DT |

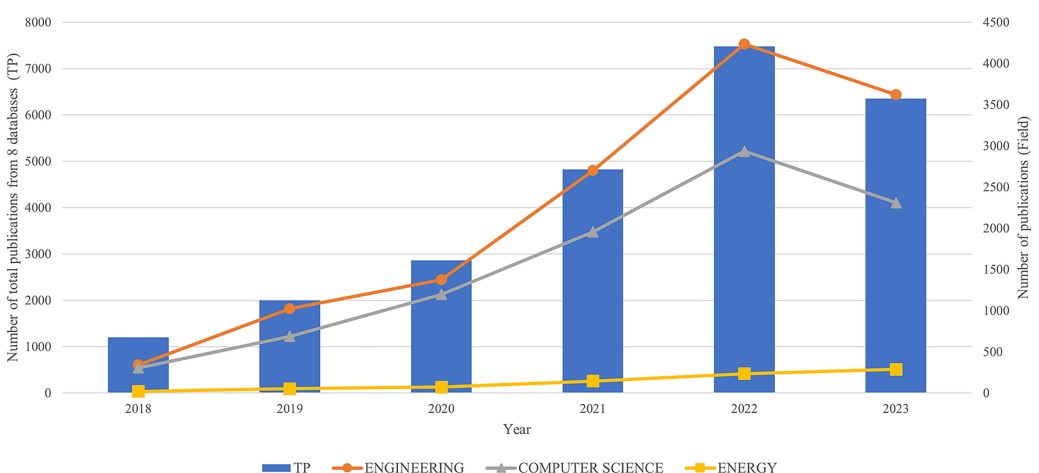

**Figure 1 Trend publications by year from 2018 to 2023.**

observed. Three prominent disciplines, namely engineering, computer science, and energy, have had a consistent upward trend in growth. From 2018 to 2023, the total publications (TP) from eight databases amounted to 24,724 for all PdM research fields using DT. Of these 24,724 publications, the three fields with the most publications were selected, namely 13,287 publications in engineering, followed by 9,391 publications in computer science and 809 in energy. Note that the total publications for other fields accounted for 1,237.

### Identification

The process of identifying and selecting relevant studies consisted of four major steps. First, the keywords for each topic area were identified. Thesauruses, encyclopaedias, and prior investigations were utilised in order to discover suitable key phrases. Second, as provided in Table 3, in November 2023, search algorithms were developed using the keywords and characteristics of the eight databases, depending on the title of the articles. Numerous articles from various sites covering related search terms might be found. Among the keywords utilised in the choosing process are "equipment", "machine", "maintenance", "smart maintenance", "predictive", "digital twin", and "virtual model".

Furthermore, this review emphasises the research publications to evaluate the methodology, data, and conclusions of the original studies that align with the purpose of this review. As a result, 278 articles from Scopus and 164 articles from Web of Science were discovered, and 5,242 items were determined in the six additional databases. Science

**Table 3  Database search strings.**

| Searching texts/search strings | Web of Science | Scopus | IEEE Xplore | Medline complete | Emerald | Springer link | Dimensions | Science direct |
|---|---|---|---|---|---|---|---|---|
| (("Equipment*" OR "machine*") AND ("maintenance*" OR "smart maintenance*") AND ("predict*") AND ("digital twin*" OR "virtual model*")) | 188 | 371 | 409 | 736 | 2,000 | None | 83 | None |
| Equipment AND maintenance AND predictive AND digital twin | 53 | 106 | 107 | 864 | 247 | 7,907 | 111 | 4,336 |
| Machine AND smart maintenance AND predictive AND virtual model | 10 | 8 | 28 | 1,269 | 649 | 13,917 | 14 | 7,713 |
| Equipment AND smart maintenance AND predictive AND virtual model | 5 | 5 | 11 | 1,023 | 498 | 11,889 | 8 | 6,326 |
| Machine AND maintenance AND predictive AND digital twin | 120 | 200 | 187 | 963 | 295 | 8,803 | 200 | 4,923 |
| Predictive maintenance AND digital twin | 302 | 511 | 397 | 1,543 | 287 | 10,079 | 521 | 7,249 |
| Total number of duplicates | 490 | 830 | 730 | 5,662 | 1,976 | 52,595 | 854 | 30,547 |
| Subtotals containing duplicates | 93,684 | | | | | | | |
| Total selected articles | 34 | | | | | | | |

Direct, IEEE Xplore, Emerald, Medline Complete, Dimensions, and Springer Link are the other six databases. Other methods, such as websites, organisations, and citation searches, were used to find pertinent studies. For the other methods of this study, the authors use the website Google Scholar. It is one of the other alternatives as a search engine, in addition to the eight databases used in this study. A total of 5,699 references, including articles and reports, were obtained after fifteen references were discovered by utilising similar keywords.

A software programme called VOSviewer is used to make maps from network data and to visualise and explore those maps. The VOSviewer produces a cluster, a collection of items contained within a map, while a network is a collection of items and the connections among the items in a map. Other than that, different colours denote the collection of items on a map and are used to distinguish the clusters. The boxes in the maps represent the object of interest, and the connections between two distinct objects are indicated by the link in the connection. A line's width denotes the strength of a link, while a longer length denotes the availability of more publications online.

The visualisation map exhibits the terms or keywords present in the data file, utilising the clustering methodologies accessible in VOSviewer. The strategy is founded upon mapping and clustering, both of which operate on the identical fundamental principle (*Bukar et al., 2023*; *Waltman, van Eck & Noyons, 2010*). By examining the connections between the keywords in a variety of publications, co-occurrence analysis aims to identify timely issues and assist scholars in comprehending contemporary scientific concerns (*Yin et al., 2022*).

The crucial cluster or strong connection is indicated in red in Fig. 2, which provides an overview of PdM and DT research's six distinct association clusters. The colour scheme is used to identify six distinct clusters: red denotes seven study subjects, green and blue for six

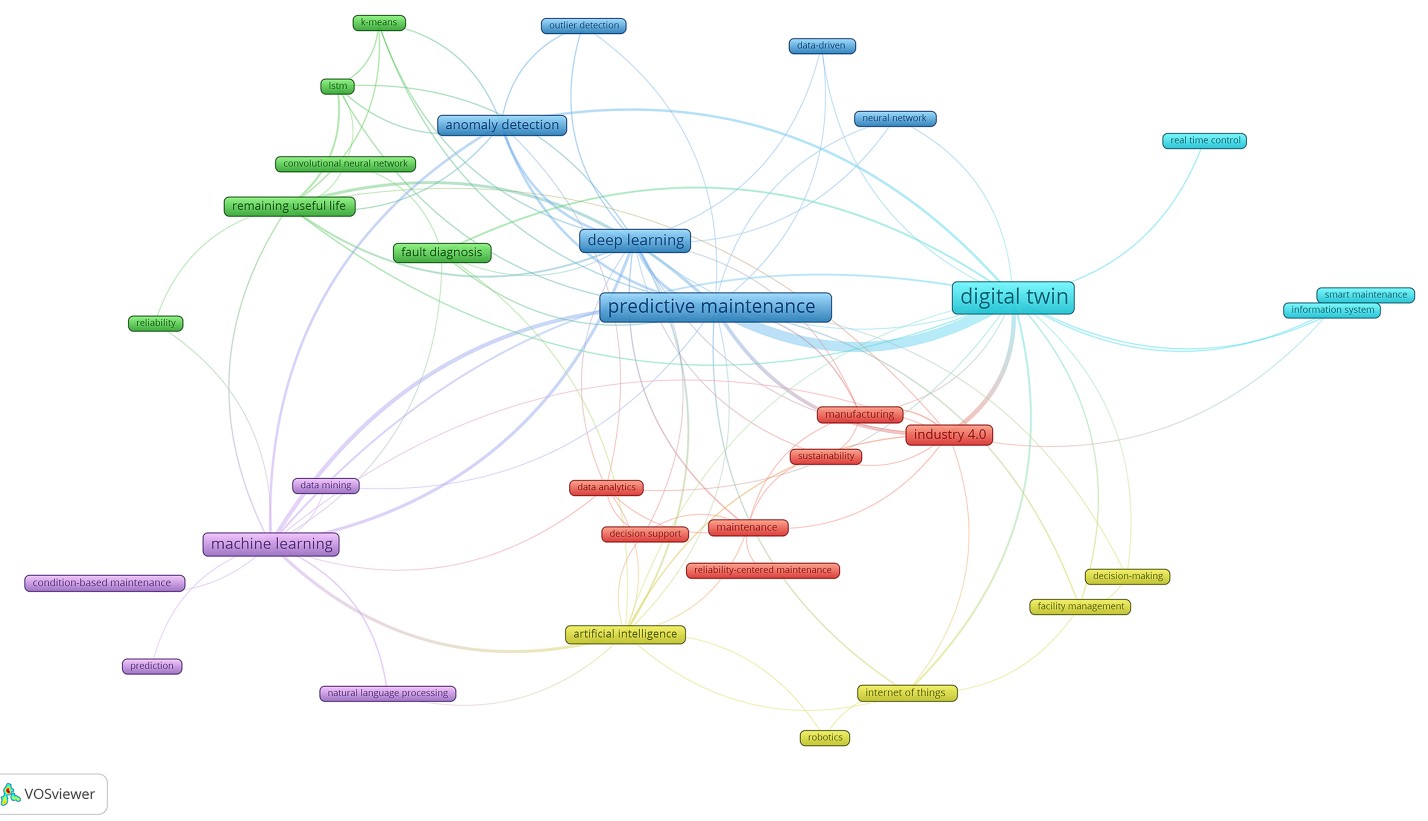

**Figure 2 A general overview of PdM using DT and their relationship and connection to each other.**

topics, purple and yellow for five topics, and light blue for the smallest cluster, which has four themes. The primary subjects covered by the largest cluster in Fig. 2 include data analytics, Industry 4.0, maintenance, decision support, *etc*. Thus, these phrases will be used as keywords in this study's investigation of these crucial subjects.

In Fig. 3 for the DT study, Cluster 1 in red consists of three items: data analytics, Industry 4.0, and manufacturing. Meanwhile, Cluster 2 in the purple colour indicates one item of ML. The following group of clusters, namely Cluster 3 in the colour yellow, with four different items, comprises decision-making, facility management, the Internet of Things, and artificial intelligence. Cluster 4 in light blue represents three items: real-time control, smart maintenance, and an information system. Cluster 5 in blue stands for five items: predictive maintenance, deep learning, anomaly detection, data-driven systems, and neural networks. Finally, Cluster 6 in green stands for two items: real-time fault diagnosis and remaining useful life.

The two items in Cluster 1 (highlighted in red) of Fig. 4 for the PdM study are manufacturing and Industry 4.0. Consequently, two items are indicated by Cluster 2, which is coloured purple: data mining and machine learning. Artificial intelligence, facility management, the Internet of Things, and decision-making are the four items in Cluster 3, which is coloured yellow. The single point represented by Cluster 4 in the colour light blue
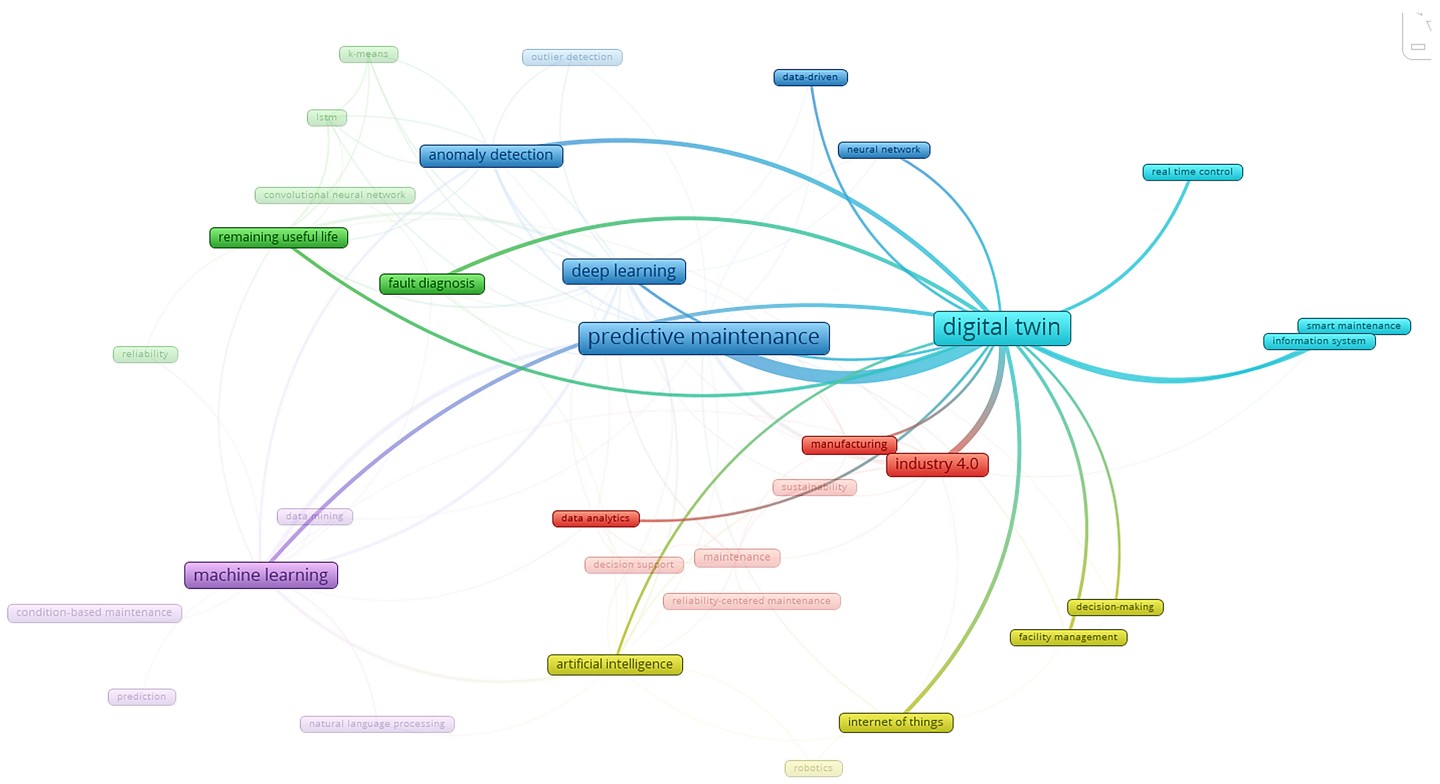

**Figure 3** The connection between DT and key regions in different clusters.

is the digital twin. On the other hand, six items depicted by Cluster 5 in blue are neural networks, data-driven, outlier detection, anomaly detection, and deep learning. Lastly, the four items represented by Cluster 6 in green are fault diagnosis, remaining useful life, LSTM, and k-means.

These results indicate a lot of information mapped under the PdM and DT. All these important details on a wide range of subjects available for thorough comprehension will be covered in more detail in the results section.

### Screening

Among 5,699 references, 5,684 were obtained using databases and registries, while 15 were discovered using databases and other methods (see Fig. 5). Duplicate references and unrelated topics were separated from the obtained references. The databases contained 387 duplicate articles, and the other approaches yielded ten referrals. After deleting redundant references, 5,297 articles for the databases and five for the other approaches remained. Duplicate references were eliminated, and the remaining references were then carefully inspected by observing the title, keywords, and abstract.

Some criteria were also considered in other ways. First, the title and keywords included the general terms PdM, DT, and equipment. Second, a reference to the quantitative method for evaluating PdM performance using DT. From the 5,297 articles in the databases, 5,249 were removed due to inappropriate content to the themes, leaving 48

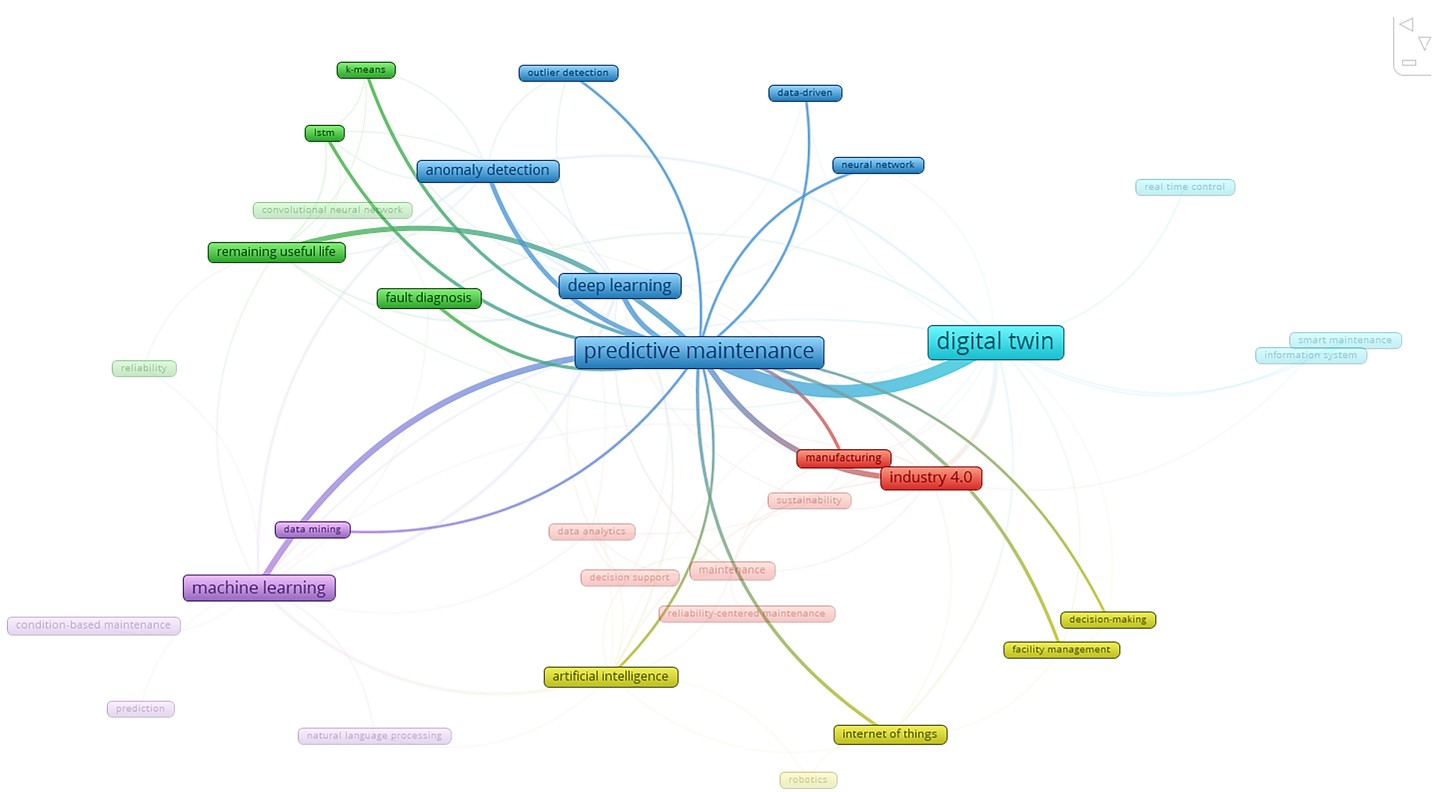

**Figure 4** The connection between PdM and key regions in different clusters.

articles pertinent to the study. However, 16 of these 48 articles were eliminated due to inaccessible full-text articles in databases, and three of five were excluded using other methods. As a result, only 32 articles from databases and two reports on other approaches were discovered and chosen for the next phase.

### Eligibility

The whole text of the publications was reviewed in this step to ensure that the 32 research articles and two reports were synthesised and analysed. The articles' essential material was thoroughly evaluated to meet the inclusion and exclusion criteria. The study's goal, methods, and research conclusions were all thoroughly assessed. As a result, no database articles or descriptions of other methodologies were included in these investigations. A hand search resulted in the inclusion of two additional pertinent articles. Consequently, 34 additional publications were included in this analysis.

### Quality evaluation and data extraction

To ensure the publications' suitability for research, the authors conducted a quality assessment of the ones they had chosen. The remaining articles were assessed using the qualitative analysis method. The research purpose, research technique, contribution, and article highlights were among the criteria used to evaluate the publications. The authors also used the information they excluded from the publications for additional synthesis and

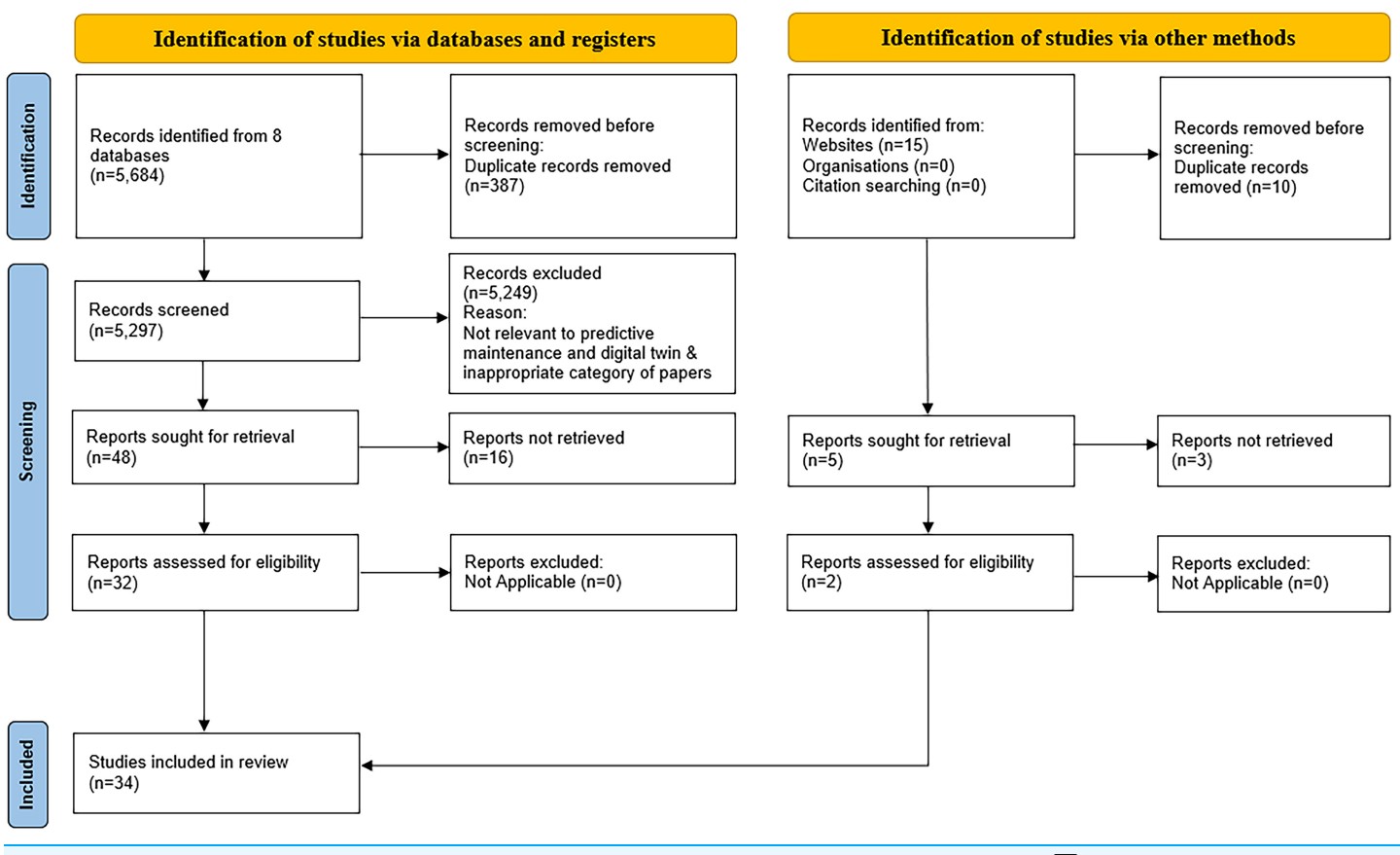

**Figure 5  The study's PRISMA flowchart.**               

analysis. An organised table was created to compile the retrieved data. The authors then went over all the synthesised data. The synthesised data were divided into categories using thematic analysis (*Braun & Clarke, 2006*; *Guest, Namey & Chen, 2020*; *Nowell et al., 2017*). The authors extensively explored the classification of the input properties. The authors reached a consensus on any differences or contradictions prior to reaching an agreement.

## RESULTS

### Background of the selected studies

The majority of the articles included in this analysis (nine) originated from China, as shown in Fig. 6. In addition, three studies were conducted in India, and one reported study was conducted in Malaysia and Singapore, bringing the total number of studies conducted in Asia to 14. Sixteen additional studies were conducted in Europe, including three in Norway and four in Greece and Italy. Meanwhile, two studies were conducted in the United Kingdom and Germany, as well as one in France. Two studies were conducted in North America, including Canada and the United States. One study was conducted in Australia, and one was conducted in Turkey.

In terms of the sectors of the selected articles, the PdM and DT integration was spearheaded and majorly involved in mechanical engineering (manufacturing) industries.

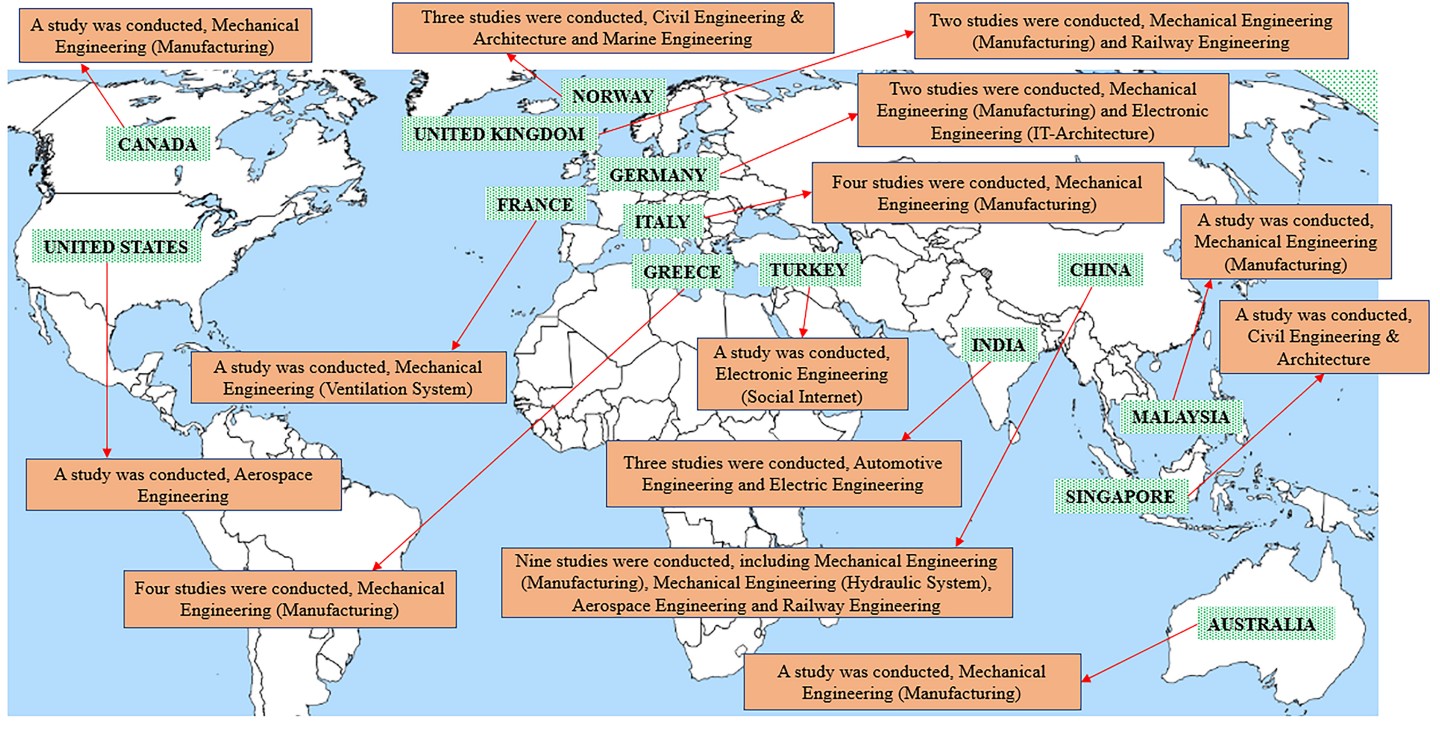

**Figure 6 Countries and industries involved in the studies.**

It was recorded that 18 studies dealing with this industry sector. Secondly, three studies in civil engineering and architecture involve building systems. Meanwhile, there are two studies for industries related to the railway, aerospace engineering, automotive engineering, and mechanical engineering (hydraulic system). This is followed by other industries, namely mechanical engineering, electronic engineering (IT-architecture), electronic engineering (social internet), electrical engineering, and marine engineering, with a specialisation in renewable energy sources.

A summary of the existing literature and the research gap on PdM with DT is presented in Table 4. By referring to the summary, the authors and year, field of interest, technique, and research strategies for PdM using DT are provided for each article. In addition, the outcomes and research gaps/future work to improve future research studies are mentioned.

In 2019, there were six articles released, and in 2020, five studies came out, five studies were published in 2021, nine articles were published in 2022, and nine studies were published in 2023.

Figure 7 demonstrates the extensive adoption of PdM using DT across various industries. Successful implementations were evident in seven of the ten sectors: automotive, civil and architectural, aerospace, mechanical, marine, railway, electric and electronic engineering. However, notably absent from current implementations are three pivotal industries: healthcare, utilities (smart water management), and agriculture (smart farm).

**Table 4 Existing literature and research gap on PdM using DT.**

| Authors, year and field | Field of interests | Technique of PdM using DT | | Predictive outcome of PdM using DT | | | | Maintenance management methods & strategies | | | | Outcome | Research gap/Future work |
|---|---|---|---|---|---|---|---|---|---|---|---|---|---|
| | | Deep learning (DL) | Machine learning (ML) | Remaining useful life (RUL) estimation | Prognostics and health management (PHM) | Anomaly detection | Fault diagnosis | Condition based maintenance (CBM) | Reliability centred maintenance (RCM) | Failure mode and effects analysis (FMEA) | Computerised maintenance management system (CMMS) | | |
| *Liu et al. (2019)* Mechanical engineering (Manufacturing) | Fault prediction and maintenance method (Mechanical equipment) | | | | ✓ | | ✓ | | | | | Results show that when convergence rates are comparable, this method's model prediction error is lower than the conventional methods. | Enhancing the three-layer super network's data model in accordance with DT's deep data qualities and creating algorithms that are more suited to the model's operation. |
| *Aivaliotis, Georgoulias & Chryssolouris (2019)* Mechanical engineering (Manufacturing) | Robots | | | ✓ | ✓ | | | | | | ✓ | Before performing a job, the user is able to determine how it will influence the machine's status. | Include the suggested approach in a larger DT using the PdM framework, whose major objective will be to evaluate the condition of the machinery and schedule maintenance procedures. |
| *Werner, Zimmermann & Lentes (2019)* Mechanical engineering (Manufacturing) | Integrated PdM approach with DT (Mechanical equipment) | ✓ | | ✓ | ✓ | | | | | | ✓ | To enhance the estimation of RUL, the DT could produce outcomes for retrofitting data-driven models for prediction. | Future work must incorporate data modelling and physics-based simulation processes. It is necessary to check and verify the presented data transfer interfaces and mathematical models with real-world instances. |
| *Rajesh et al. (2019)* Automotive engineering | System of brake pads for automobiles | ✓ | | | ✓ | ✓ | | | | | | With an average inaccuracy of 11%, the results reveal a good connection between the simulation and computed values. | The creation of an entire DT of the vehicle with the assistance of more sensors and subsystems will be the main goal of future work. |
| *Aivaliotis et al. (2019)* Mechanical Engineering (Manufacturing) | Robots | | | ✓ | ✓ | | | | | | | The model was simulated, and the results of the simulation signals were compared to those measured by the actual robot. Multiple iterations of the simulated comparing refining process were conducted up until the intended outcome was attained. | Future work will involve replicating the process for another type of equipment and validating it in an operational plant. Furthermore, the accuracy of the developed models will be enhanced in order to remove discrepancies between the actual data and the projected performance. |

(Continued)

| Authors, year and field | Field of interests | Technique of PdM using DT | | Predictive outcome of PdM using DT | | | | Maintenance management methods & strategies | | | | Outcome | Research gap/Future work |
|---|---|---|---|---|---|---|---|---|---|---|---|---|---|
| | | Deep learning (DL) | Machine learning (ML) | Remaining useful life (RUL) estimation | Prognostics and health management (PHM) | Anomaly detection | Fault diagnosis | Condition based maintenance (CBM) | Reliability centred maintenance (RCM) | Failure mode and effects analysis (FMEA) | Computerised maintenance management system (CMMS) | | |
| *Altun & Tavli (2019)* Electronic Engineering (Social Internet) | PdM application to commercial devices *via* distributed thing-to-thing communication. | ✓ | ✓ | | | ✓ | | | | | | A new model of reference that applies PdM to socially communicating domestic appliance DTs. Examines how PdM is applied using the original model. | Not specified in the research study. |
| *Luo et al. (2020)* Mechanical Engineering (Manufacturing) | CNC Machine Tool (CNCMT) driven | ✓ | ✓ | | ✓ | | ✓ | | | | | In comparison to a single strategy method, a hybrid PdM approach enables a better DT model and data integration and application, which can lead to more accurate results. | Model migration learning based on DT, as well as Cloud and Edge-based DT model implementations. |
| *Mi et al. (2020)* Mechanical Engineering (Manufacturing) | Bearings for vertical mill | | | | ✓ | | ✓ | | | | | To demonstrate the potential and superiority of the proposed technique, an actual technical scenario is examined. | Future studies will provide practical technical assistance in this area. |
| *Heim et al. (2020)* Aerospace engineering | Aeroplane maintenance parts | ✓ | ✓ | ✓ | | | | | | | | The preliminary findings revealed that, despite the addition of the extra qualitative parameters of the level of procedure and kind of situation. The precision for this method was 88% on average for all parts tested. | To provide a more complete and in-depth look into the stresses in particular regions of the aircraft, future work will involve the propagation of cracks in the aircraft frame. |
| *Rossini et al. (2020)* Mechanical engineering (Manufacturing) | An adaptable and scalable structure for smart DT implementation (Mechanical equipment) | | ✓ | | ✓ | | ✓ | | | | | When compared to additional traditional ways, utilising a DT for PdM can produce superior outcomes. | The PdM module includes an optimisation module for scheduling predicted maintenance tasks according to current planning and a prediction module for future failures of components. |
| *Centomo, Dall'Ora & Fummi (2020)* Mechanical engineering (Manufacturing) | Approach for the automation of electronic development (Mechanical equipment) | | | | ✓ | ✓ | ✓ | | | | | The total number of recipes increases in proportion to a given number of Predictive Maintenance Supervisor (PMS) states. Furthermore, implementing the PMS and thresholds takes a shorter time and has a low complexity with just one Monitoring State Machine (MSM). | In upcoming work, the author will assist in developing new standards. |

| Authors, year and field | Field of interests | Technique of PdM using DT | | Predictive outcome of PdM using DT | | | | Maintenance management methods & strategies | | | | Outcome | Research gap/Future work |
|---|---|---|---|---|---|---|---|---|---|---|---|---|---|
| | | Deep learning (DL) | Machine learning (ML) | Remaining useful life (RUL) estimation | Prognostics and health management (PHM) | Anomaly detection | Fault diagnosis | Condition based maintenance (CBM) | Reliability centred maintenance (RCM) | Failure mode and effects analysis (FMEA) | Computerised maintenance management system (CMMS) | | |
| *Das et al. (2021)* Automotive Engineering | Li-ion battery packs for a fleet of vehicles | | ✓ | | ✓ | ✓ | | | | | | The outcomes establish the State of Health (SOH) prediction system, the anomaly detection system, and the efficacy of the suggested model, which NASA has examined. | The framework can be expanded to enable various use cases for different stakeholders, including Original Equipment Manufacturers (OEM), service providers, and vehicle operators. |
| *Xiong et al. (2021)* Aerospace engineering | Aero-engine | ✓ | | ✓ | ✓ | | ✓ | | | | | According to the results of the experiment, the data set used to train the model is 80 percent comprehensive, the model's predictions are highly accurate, and Aeroengine RUL's estimated Root Mean Square Error (RMSE) is 13.12, which is lower than that of other experimental systems. | Long Short-Term Memory (LSTM) is a supervised learning technique. To improve the PdM of civil aviation operations safety, efficient semi-supervised and unsupervised learning techniques for fault diagnosis and RUL prediction must be investigated and embedded in the DT model. |
| *Moghadam, de Rebouças & Nejad (2021)* Marine engineering | Floating offshore wind turbine drivetrains' gearboxes | | | ✓ | ✓ | | ✓ | | | | | The calculated contact loads and stresses were in good agreement, and the method based on a linear torsion model may be used with fully automatic turbine control and is computationally quick. | The algorithm was simulated using Hardware-In-the-Loop (HIL) to see if it could be executed in real-time for failure prediction, handle the aforementioned practicality issues, and be integrated into a working wind turbine drivetrain system. |
| *Pillai, Shih & Roberts (2021)* Railway engineering | Switch and crossing (S&C) rails | | | | ✓ | | ✓ | | | | | The outcomes for predicting regions prone to surface damage. | The mechanical behaviour of the rail's substrate can be predicted using a FE model. |
| *Yang et al. (2021)* Railway engineering | Switch machines | ✓ | | | | | | | | | | Maintenance staff can create a suitable maintenance schedule by combining the switch machine's visual model and the findings of condition prediction. | The DT model can be supplied data from numerous sources of switch machines. The DT using the PdM framework is applicable to PM for switch machines as well as other pieces of equipment. |
| *Hosamo et al. (2022)* Civil engineering & Architecture | Air handling unit (AHU) (Systems used in building) | | ✓ | | | | ✓ | | | | ✓ | The findings demonstrate the effectiveness and utility of the automatic fault detection approach in AHUs. | A cutting-edge data model that establishes a standardised data integration solution for a variety of sensor types and application platforms using an ontology-based methodology. |

*(Continued)*

| Authors, year and field | Field of interests | Technique of PdM using DT | | Predictive outcome of PdM using DT | | | | Maintenance management methods & strategies | | | | Outcome | Research gap/Future work |
|---|---|---|---|---|---|---|---|---|---|---|---|---|---|
| | | Deep learning (DL) | Machine learning (ML) | Remaining useful life (RUL) estimation | Prognostics and health management (PHM) | Anomaly detection | Fault diagnosis | Condition based maintenance (CBM) | Reliability centred maintenance (RCM) | Failure mode and effects analysis (FMEA) | Computerised maintenance management system (CMMS) | | |
| *Wang et al. (2022)* Mechanical engineering (Hydraulic System) | Hydraulic system (Hydraulic Equipment) | ✓ | | | ✓ | | ✓ | | | | | The experimental findings show that, in the absence of sufficient prior defect data, the diagnostic accuracy of ten common hydraulic cylinder faults can approach 89%, which represents a 9% improvement over a non-interactive simulation model. | Given the lack of feature data in actual applications, the DT technique's ability to foresee fault characteristics and swiftly update models will have significant application value in the future. |
| *Bondoc, Tayefeh & Barari (2022)* Mechanical engineering (Manufacturing) | The vibration of a structural asset/machine (Mechanical equipment) | | | | ✓ | | ✓ | | | | | HF digital model with a precursive sensor network for health monitoring. Accuracy in this learning phase depends on the presumptions held. | Reduce the effects of obvious assumptions, like picturing a fluid tank. The system's intrinsic frequencies will be impacted by the tank's enhanced pre-stress and mass effects. |
| *Yakhni et al. (2022)* Mechanical Engineering (Ventilation System) | Condition monitoring of ventilation systems (Mechanical equipment) | | | | | | ✓ | | | | | Experimental and simulated findings show the effectiveness of this created method. | The method that has been developed can be applied to many industrial systems and issues. Processing current signals can be done using a variety of techniques. |
| *Avornu et al. (2022)* Mechanical Engineering (Manufacturing) | Data fusion approach for PdM and DT (Mechanical equipment) | ✓ | | | | ✓ | ✓ | ✓ | | | | Support Vector Machine (SVM) is the algorithm that performs the best in the presented study. | To make additional space for future research, the dataset was trained using a variety of machine-learning algorithms. |
| *Mubarak et al. (2022)* Mechanical Engineering (Manufacturing) | Open system design enabling condition-based maintenance (Mechanical equipment) | ✓ | ✓ | ✓ | | | ✓ | ✓ | | ✓ | | It is anticipated that the suggested approach will make maintenance more affordable and enhance both the predictive process's intelligence and the precision of the predicted outcomes. | Not specified in the research study. |
| *Panagou et al. (2022b)* Mechanical Engineering (Manufacturing) | Rolling mill | ✓ | | | | ✓ | ✓ | | | | | The outcomes demonstrate that the two sensors may predict real-world favorable circumstances. In order to raise confidence and dependability ratings, two of these sensors are also employed in DT scenarios and real-time conditions. | Not specified in the research study. |
| *Zhang et al. (2022)* Mechanical engineering (Hydraulic System) | Hydraulic System for Shearers | | | | | | ✓ | | | | | There are other benefits from using a Back Propagation Neural Network (BPNN) dependent on a grey rough, including smoother fitting and a smaller variation in the prediction outcomes. | Not specified in the research study. |

Abd Wahab et al. (2024), *PeerJ Comput. Sci.*, DOI 10.7717/peerj-cs.1943

| Authors, year and field | Field of interests | Technique of PdM using DT | | Predictive outcome of PdM using DT | | | | Maintenance management methods & strategies | | | | Outcome | Research gap/Future work |
|---|---|---|---|---|---|---|---|---|---|---|---|---|---|
| | | Deep learning (DL) | Machine learning (ML) | Remaining useful life (RUL) estimation | Prognostics and health management (PHM) | Anomaly detection | Fault diagnosis | Condition based maintenance (CBM) | Reliability centred maintenance (RCM) | Failure mode and effects analysis (FMEA) | Computerised maintenance management system (CMMS) | | |
| *Panagou et al. (2022a)* Mechanical engineering (Manufacturing) | Rolling mill | | ✓ | | ✓ | ✓ | ✓ | | | | | With one false positive in the confusion matrix, the training's final accuracy was 0.99. After the feature significance was filtered, two sensors had values greater than 0.05. | Not specified in the research study. |
| *Aivaliotis et al. (2023)* Mechanical engineering (Manufacturing) | Robots | | | ✓ | ✓ | ✓ | | | | | | The results of this study indicate that a DT can be designed, built, and operated with 95% accuracy. | This includes replication for a different machine type, like a CNC milling machine, validation in several industry sectors, and validation of the approach at a production facility. |
| *Singh et al. (2023)* Electric engineering | Induction motors | | ✓ | ✓ | ✓ | ✓ | ✓ | | | | | It can be inferred that the models exhibit coherence on the two software platforms, indicating that they remain acceptable with the exchange of data even with the platform transition. | A workable remedy for the appearance of both short- and long-term conditions is developed by the proposed research. |
| *Hu et al. (2023a)* Civil engineering & Architecture | Indoor climate | ✓ | ✓ | ✓ | | | ✓ | | | | | The algorithm achieves its best performance with a warning time of 30 minutes (Precision = 0.96, Accuracy = 0.87, F1 score = 0.86, and Recall = 0.77). | To increase the prediction model's precision and dependability, future studies should employ devices in various interior building locations and take other pertinent factors into consideration. |
| *Hosamo et al. (2023)* Civil engineering & Architecture | Building information modeling (BIM) and HVAC System | | ✓ | ✓ | ✓ | | | | | | ✓ | For the forecast, the optimal approach was Extreme Gradient Boosting (XGB). XGB exhibits an accuracy advantage of up to 5% over the other models and an average of 2.5% over Multi-Layer Perceptron (MLP). Compared to XGBoost, Random Forest is about 96% quicker and easier to use. | Prospective research avenues encompass investigating alternative machine learning techniques, integrating additional variables to the probability model of comfort for users, and broadening the framework's use. |
| *Feng et al. (2023)* Mechanical engineering (Manufacturing) | Offshore production system for oil and gas | | | ✓ | | | | | | | | In a reasonable amount of time, traditional variable neighborhood search (VNS) can produce decent solutions, but as the data scale grows, the difference from the optimal result rises progressively. | In order to more precisely simulate the real mechanisms of communication across numerous organisations, future research can try to develop a more comprehensive PdM decision-making model based on DT. |

*(Continued)*

| Authors, year and field | Field of interests | Technique of PdM using DT | | Predictive outcome of PdM using DT | | | | Maintenance management methods & strategies | | | | Outcome | Research gap/Future work |
|---|---|---|---|---|---|---|---|---|---|---|---|---|---|
| | | Deep learning (DL) | Machine learning (ML) | Remaining useful life (RUL) estimation | Prognostics and health management (PHM) | Anomaly detection | Fault diagnosis | Condition based maintenance (CBM) | Reliability centred maintenance (RCM) | Failure mode and effects analysis (FMEA) | Computerised maintenance management system (CMMS) | | |
| *Mourtzis, Tsoubou & Angelopoulos (2023)* Mechanical engineering (Manufacturing) | Robots | | ✓ | ✓ | ✓ | | | | | ✓ | | The evaluated ML model predicts "urgent" data with a 100% accuracy rate, while "good" and "alert" data are predicted accurately 96% and 96.4% of the time, respectively. | The authors envisage a variety of robot configurations for the purpose of future studies, including hybrid cells that feature human operators coexisting within the cell and collaborating with robotic arms. |
| *Siddiqui, Kahandawa & Hewawasam (2023)* Mechanical engineering (Manufacturing) | Automation systems (Mechanical equipment) | ✓ | ✓ | | | ✓ | | | | | | The outcomes showed the DT created in this study could successfully identify anomalies in the automated system since the trained model functioned incredibly well. | Not specified in the research study. |
| *Harries et al. (2023)* Mechanical engineering (Manufacturing) | RUL of the bicycle factory's machinery | | ✓ | ✓ | ✓ | ✓ | | ✓ | | | | PdM outperformed Time-Based Maintenance (TBM) on average when it came to degradation rather than similarities. For linear profiles, the similarity model outperformed TBM by 8.1%, while for exponential profiles, it outperformed TBM by 4.8%. | Not specified in the research study. |
| *Mrzyk et al. (2023)* Electronic engineering (IT-Architecture) | Flexible IT-architecture | | | | ✓ | | | ✓ | | | | This evaluation's result demonstrates that the DT for PdM can be grouped into characteristic functioning, which can then be further subdivided into tool-specific, standardised, and used particularly to the case elements. | Building on these findings, future research might examine how to further minimise the resources needed to produce DT and whether these findings can be applied to additional tool solutions found in the IT framework. |

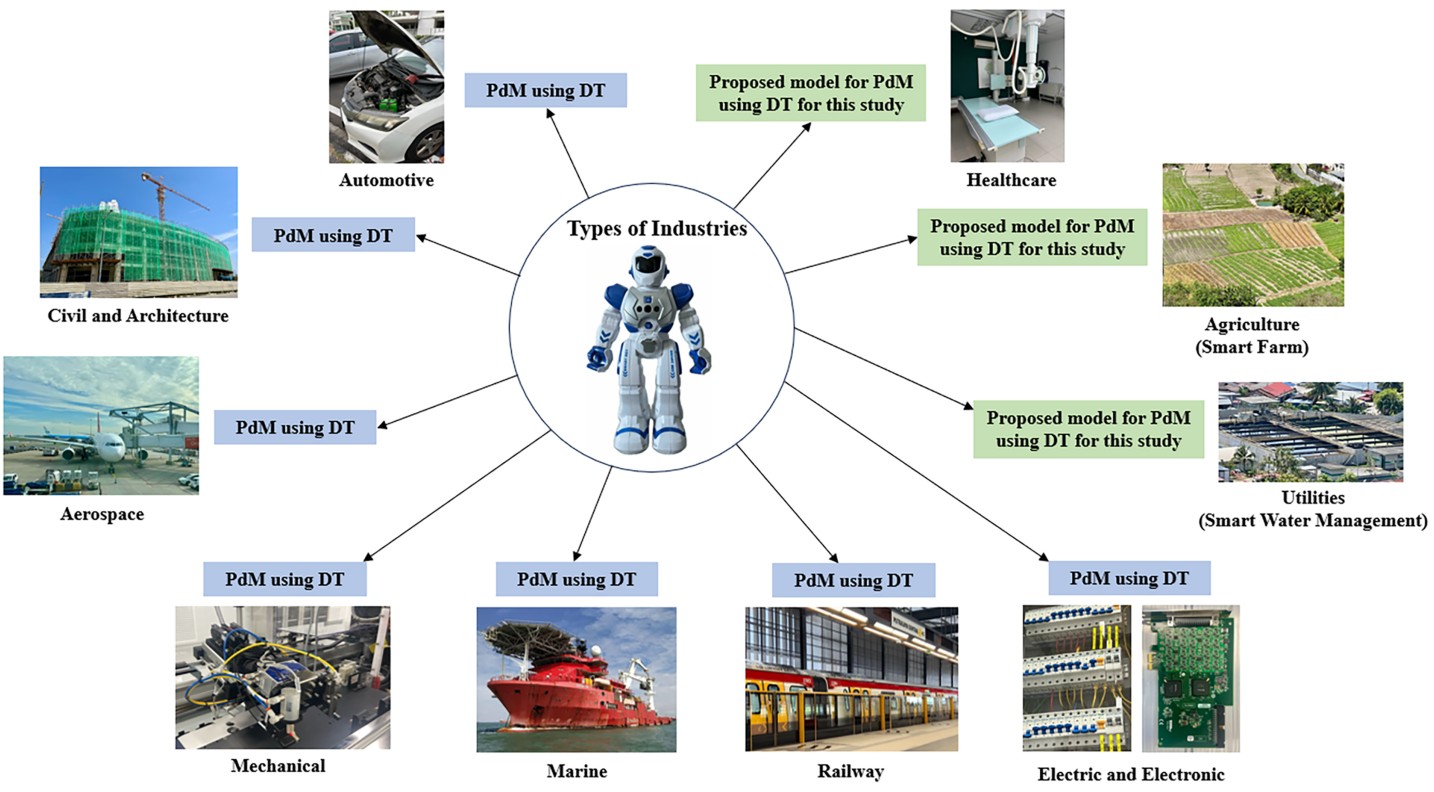

**Figure 7  Types of industries.** Photo credit: Nur Haninie Abd Wahab.

The deliberate selection of these industries (healthcare, utilities (smart water management), and agriculture (smart farm)) derives from their underexplored status in the field of PdM using DT. Despite their crucial significance, these sectors remain largely untapped in terms of harnessing the synergies of PdM and DT technologies. The prioritisation of these sectors is based on both their relevance and their effect. Integrating PdM and DT in healthcare may bring about a revolutionary improvement in equipment dependability. This integration ensures that the equipment consistently and optimally functions, which is crucial for providing high-quality treatment when patient's well-being and lives are at risk. Likewise, in the fields of utilities (smart water management) and agriculture (smart farm), this connection is crucial to reduce periods of inactivity, ensuring the dependability of systems vital to everyday life.

## Main studies outcomes

The results were obtained from the thematic analysis conducted on the selected articles. Thematic analysis was performed on 34 selected articles and revealed four main themes: (1) field of interest and types of models; (2) approaches; (3) predictive outcome; and (4) implementation of maintenance management. The following subsections describe the background of the selected research.

## Field of interest and types of models

The fundamental concept of PdM using DT was applied in various applications such as a battery for vehicles (*Das et al., 2021*), CNC machine tool (CNCMT) (*Luo et al., 2020*), aero-engine (*Xiong et al., 2021*), railways (*Pillai, Shih & Roberts, 2021*; *Yang et al., 2021*), hydraulic equipment and system (*Wang et al., 2022*; *Zhang et al., 2022*), robots, (*Aivaliotis et al., 2023*; *Aivaliotis, Georgoulias & Chryssolouris, 2019*; *Aivaliotis et al., 2019*; *Mourtzis, Tsoubou & Angelopoulos, 2023*), ventilation systems (*Yakhni et al., 2022*) and others. Several types of models can be discovered in the results of previous studies related to the field of interest. These models include the physical model (PM), the behavioural, the decision-making, and the hybrid model.

### Physical model

A PM is used to simulate physical qualities and loads (*van Dinter, Tekinerdogan & Catal, 2022*). The PMs are descriptive models. Descriptive models are primarily used for comprehension, prediction, and communication (*Kaul, Bender & Sextro, 2019*). According to Fig. 8, the percentage of PM in previous studies was 41%. The 14 studies involved in PM are *Aivaliotis et al. (2023)*, *Aivaliotis, Georgoulias & Chryssolouris (2019)*, *Aivaliotis et al. (2019)*, *Altun & Tavli (2019)*, *Avornu et al. (2022)*, *Centomo, Dall'Ora & Fummi (2020)*, *Heim et al. (2020)*, *Liu et al. (2019)*, *Mubarak et al. (2022)*, *Rajesh et al. (2019)*, *Rossini et al. (2020)*, *Siddiqui, Kahandawa & Hewawasam (2023)*, *Singh et al. (2023)* and *Yakhni et al. (2022)*. PMs also have drawbacks, such as being time-consuming and expensive to destroy and restore.

The studies conducted by *Aivaliotis et al. (2023)*, *Aivaliotis, Georgoulias & Chryssolouris (2019)*, *Aivaliotis et al. (2019)* focused on forecasting the remaining useful life (RUL) of machines and methodology to enable dynamic DT and virtual model evolution in industrial robotics were studied. *Yakhni et al. (2022)* utilised the DT approach to monitor ventilation system conditions by incorporating fault prediction and maintenance methods (*Liu et al., 2019*).

Notably, this study indicated that eight studies applied ML techniques for PdM through data-driven modelling across various sectors such as aeroplane maintenance, automobile brake pads, and induction motors (*Altun & Tavli, 2019*; *Avornu et al., 2022*; *Heim et al., 2020*; *Mubarak et al., 2022*; *Rajesh et al., 2019*; *Rossini et al., 2020*; *Siddiqui, Kahandawa & Hewawasam, 2023*; *Singh et al., 2023*). Additionally, performance prediction studies have demonstrated the potential of ML as a useful technique.

### Behavioural model

The behavioural model elucidates how external stimuli, such as driving forces or disruptive influences, impact the physical system (*van Dinter, Tekinerdogan & Catal, 2022*). These models serve a dual purpose, serving as descriptive models while enabling a comprehensive understanding of system component deterioration and their dynamic behaviour, specifically in reliability modelling. It is based on industry-standard models, such as Systems Modelling Language (SysML), Matrix laboratory (Matlab)/Simulation and link (Simulink), Dynamic modelling laboratory (Dymola), and MSC Automated Dynamic

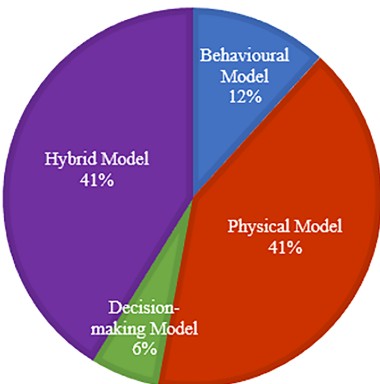

| Type of Model | Total Field of Interest |
|---|---|
| Physical Model | 14 |
| Behavioural Model | 4 |
| Decision-making Model | 2 |
| Hybrid Model | 14 |

**Figure 8** The field of interest and type of model representation for PdM using DT for each study.

Analysis of Mechanical Systems (ADAMS), used in model-based development processes (*Kaul, Bender & Sextro, 2019*). In the context of behavioural models, as indicated in the previous studies (*Bondoc, Tayefeh & Barari, 2022*; *Moghadam, de Rebouças & Nejad, 2021*; *Mourtzis, Tsoubou & Angelopoulos, 2023*; *Pillai, Shih & Roberts, 2021*), these studies constitute approximately 12% of the field. However, an inherent limitation of these models is their reliance on inadequate theory, which occasionally leads to inaccuracies in behaviour prediction.

Within this field of interest, *Mourtzis, Tsoubou & Angelopoulos (2023)* elaborated on ways to identify and categorise the crucial component's malfunctioning behaviour and for mechanical equipment vibration of a structural asset/machine (*Bondoc, Tayefeh & Barari, 2022*). For railway engineering, the field of interest of the behavioural model is switches and crossings (S&C) (*Pillai, Shih & Roberts, 2021*). This equipment uses PdM with DT using a proprietary DT model to improve the prediction accuracy of the results. In addition, PdM and DT, concerning marine engineering (*Moghadam, de Rebouças & Nejad, 2021*), gearboxes for floating offshore wind turbines.

### Decision-making model

The decision-making model serves as a framework for the analysis, justification, and validation of the model, incorporating an algorithm, a defined set of restrictions and guidelines, and adjustable input. An intelligent data-driven modelling technique is the decision-making model. Decision-making techniques, such as ML algorithms, lower the time-to-market of DT while increasing accuracy slightly (*van Dinter, Tekinerdogan & Catal, 2022*). Despite its significance, studies employing decision-making models for DT development are relatively scarce, comprising only 6% (*Feng et al., 2023*; *Mi et al., 2020*). Meanwhile, the field of interest for the decision-making model for mechanical equipment and bearings for vertical mills was discussed in *Mi et al. (2020)*. Additionally, *Feng et al. (2023)* highlighted the utilisation of DT in supporting multi-stage PdM for intelligent industrial systems, emphasizing its pivotal role in enhancing system reliability and efficiency.

### Hybrid model

A hybrid model is another sort of representation. A PM and a data-driven model (DDM), for example, are two models that this model combines. DDMs can use this data to discover hidden patterns that can subsequently be used to model a physical asset. According to *Errandonea, Beltrán & Arrizabalaga (2020)*, DT provides the most accurate synthetic data when employing a model-based approach. The PdM hybrid technique, superior to the single-strategy approach at each stage, can produce a more accurate RUL forecast with fewer errors (*Luo et al., 2020*). Several model representations of PdM using DT were developed. In total, 14 studies used such a hybrid model (*Das et al., 2021*; *Harries et al., 2023*; *Hosamo et al., 2022*, *2023*; *Hu et al., 2023a*; *Luo et al., 2020*; *Mrzyk et al., 2023*; *Panagou et al., 2022a*, *2022b*; *Wang et al., 2022*; *Werner, Zimmermann & Lentes, 2019*; *Xiong et al., 2021*; *Yang et al., 2021*; *Zhang et al., 2022*) with a percentage of 41% from these studies.

For the field of interest, CNC machine tools (CNCMT) (*Luo et al., 2020*) and hydraulic systems for hydraulic equipment (*Wang et al., 2022*; *Zhang et al., 2022*) offer a more practical solution method for integrated hydraulic systems for PdM using DT, made up of numerous interconnected parts and components. Meanwhile, *Hosamo et al. (2022)* state that the heart of a building's heating, ventilation, and air conditioning (HVAC) systems is its air handling unit (AHU) system. AHU stability is essential for maintaining high efficiency and increasing the lifespan of HVAC systems. For railway engineering (*Yang et al., 2021*), a separate model DT is used for switch drives to enhance the outcomes' forecasting accuracy. RUL estimation is used in seven studies for this hybrid model, which is of interest for various domains, such as flexible IT architectures, indoor climate, and others (*Harries et al., 2023*; *Hosamo et al., 2023*; *Hu et al., 2023a*; *Luo et al., 2020*; *Mrzyk et al., 2023*; *Werner, Zimmermann & Lentes, 2019*; *Xiong et al., 2021*). Other studies involving PdM and DT are in aerospace engineering, *i.e.*, aircraft aero-engine (*Xiong et al., 2021*), marine engineering (*Das et al., 2021*), rolling mills (*Panagou et al., 2022a*, *2022b*), and lithium-ion batteries for a fleet of vehicles.

## Approaches

Table 4 shows two main approaches that were frequently used: (i) deep learning and (ii) machine learning.

### Deep learning technique

DL algorithms are used to represent DT or for predictive analytics. DL is an important subset of AI and a subset of ML that demonstrates how machines can mimic humans in learning certain types of information and accumulating situational awareness (*Akhtar et al., 2023*; *Brownlee, 2017*; *Huang et al., 2023*; *Mehrjardi et al., 2023*; *Rastall & Green, 2022*; *Tanveer et al., 2023*; *Usuga-Cadavid et al., 2022*; *Velichko et al., 2023*; *Zhao et al., 2020a*; *Zhou et al., 2023*). DL techniques such as long short-term memory (LSTM), deep neural network (DNN), and other DL architectures were used (*Neo et al., 2022*). DNNs are formed by stacking these layered networks (*Brownlee, 2017*). In the review, four studies focus on DL, such as the LSTM approach (*Hu et al., 2023a*; *Xiong et al., 2021*; *Yang et al.,*

*2021*). The issue of a series' long-term reliance can be resolved with LSTM by analysing the inputs in time a series (*Li, Wang & Li, 2022*).

An example of a combined approach can be discovered in the study (*Kamat, Sugandhi & Kumar, 2021*). This study uses a combination of RUL, DL, and anomaly detection approaches. Machine supervisors must determine the RUL of rotating machinery based on deterioration data. DL models are popular and reliable techniques for predicting the breakdown of rotating machinery, such as bearings. Notably, RUL is not clearly defined when the machine is operating normally. This study suggests that anomalies be kept an eye on throughout RUL estimator training and used to overcome this problem. In addition, raw bearing vibration data are processed to extract relevant time-domain data, which is then processed using DL algorithms to look for anomalies. Consequently, this data triggers data-driven RUL estimation. Unsupervised clustering is utilised for anomaly trend analysis, and semi-supervised methods are employed for anomaly detection and RUL estimation. Estimating the equipment's RUL and detecting anomalies are improved with DL-based methods (*Kamat, Sugandhi & Kumar, 2021*; *Lin & Tao, 2019*; *Pang et al., 2021*; *Wen et al., 2021*). DL algorithms have recently been proposed as a more effective approach for extracting text features for the classification of texts (*Zhong et al., 2020*). To automatically categorise occupational injuries, *Zhang (2022)* developed a structure-based DNN with a Word2Vec model. With an F1 score of 72%, the suggested hybrid architecture demonstrated good predictive power.

### Machine learning technique

There are two types of ML, namely supervised and unsupervised learning. Supervised approaches, which may be used for both classification and regression issues, train a model based on known input and output data to predict future events. Unsupervised learning is typically applied to clustering issues and looks for hidden patterns or intrinsic structures in the input data (*Abd-Elrazek et al., 2021*). ML techniques such as naive Bayes, random forest, support vector machine (SVM), decision tree, and regression models are the most widely used AI approaches (*Khairuddin et al., 2022*, *2023*; *Neo et al., 2022*), artificial neural network (ANN), and decision trees (*Hosamo et al., 2022*) and ML is also used for Autoregressive Integrated Moving Average (ARIMA) (*Yang et al., 2021*). From the review, 19 studies use ML approaches for PdM using DT (*Altun & Tavli, 2019*; *Avornu et al., 2022*; *Das et al., 2021*; *Harries et al., 2023*; *Heim et al., 2020*; *Hosamo et al., 2022*, *2023*; *Hu et al., 2023a*; *Luo et al., 2020*; *Mourtzis, Tsoubou & Angelopoulos, 2023*; *Mubarak et al., 2022*; *Panagou et al., 2022a*, *2022b*; *Rajesh et al., 2019*; *Rossini et al., 2020*; *Siddiqui, Kahandawa & Hewawasam, 2023*; *Singh et al., 2023*; *Wang et al., 2022*; *Werner, Zimmermann & Lentes, 2019*).

According to a preliminary study conducted by *Wang & Wu (2023)*, the model extends the usual ML used in genetic studies to enable the discovery of epistatic effects. In Genome-Wide Association Studies (GWAS), it is the standard statistical method for estimating Single Nucleotide Polymorphism (SNP) interactions. With an accuracy of 99.42% for the

maintenance preventative model and 99.80% for the replacement prioritisation model, SVM performs better than other algorithms when prioritising medical devices. The prioritisation mode was created by combining supervised and unsupervised ML techniques (*Zamzam et al., 2021b*).

## Predictive outcome

The predictive results of PdM using DT are indicated in Table 4. The extracted predictive outcomes were divided into the following categories: (1) Prognostics and Health Management (PHM), (2) fault diagnosis, (3) anomaly detection, and (4) RUL estimation.

To employ DT for more effective maintenance, practically all investigations aim to estimate, foresee, or detect the status of a system, component, or system of systems. As a result, the expected result is different. These results can be divided into several predictive outcomes. In 23 studies, an attempt was made to predict PHM (health indicator). Note that 17 studies targeted fault diagnosis, and 12 targeted anomaly detection for PdM using DT. Subsequently, 17 studies aimed to estimate the RUL. In the following sections, the predictive outcomes for the predictive algorithm of PdM are described using DT.

### Prognostics and health management

According to a preliminary study conducted by *Toothman et al. (2023)*, PHM systems play a crucial role in diagnosing equipment health and anticipating potential issues. These systems offer valuable insights into the components likely to fail, expediting early detection and maintenance. Their efficacy lies in swiftly pinpointing machine parts requiring immediate attention, enhancing failure prediction. PHM aims to provide methods and tools for developing an appropriate maintenance policy for a given asset under its operating and degrading conditions to achieve high availability at the lowest possible cost. It can be perceived as a comprehensive approach to effective and efficient system health management (*Fink et al., 2020*). Utilising sensor technology and analytical capabilities, PHM continuously monitors machine health, detects degradation, and facilitates maintenance planning for various components. Sensors strategically placed on machines gather crucial raw data reflecting their condition, forming the backbone of the PHM system. It is crucial to emphasise the meticulous placement of these sensors to ensure accurate data collection. The collected data undergoes comprehensive analysis within PHM systems, enabling a deep understanding of the manufacturing system's behaviour and the proactive anticipation of potential operational challenges, as elucidated by *Ardila et al. (2020)*.

### Fault diagnosis

In studies verified using data from controlled experiments, test beds, or numerical simulations, PdM confronts various failure diagnosis and prognosis concerns that are typically overlooked (*Fernandes, Corchado & Marreiros, 2022*). Correspondingly, *Xu et al. (2019)* proposed a fault diagnosis method that uses DT technology to move fault information from a virtual entity to its physical counterpart. Therefore, fault detection and diagnostics significantly reduce downtime and unexpected failure of increasingly sophisticated industrial machinery and equipment (*Sun et al., 2017*). In contrast to

traditional fault diagnosis, which relies on engineers' extensive human expertise to determine the relationship between monitored data and machine health, intelligent fault diagnosis applies ML theories to fault diagnosis, automating the fault detection and classification process (*Lei et al., 2020*).

### Anomaly detection

Anomaly detection is the process of identifying significant deviations from the rest of the data (*López et al., 2023*; *Wang et al., 2021*). Failures can be made, and the operator is notified in time to save the system from failing by spotting anomalies (*He et al., 2023*). Thus, extensive study has gone into the field of log anomaly detection, which attempts to automatically identify any aberrant logs from log data to allow operators to swiftly address problems and ensure system stability (*Bertero et al., 2017*; *Zhao et al., 2021*). A machine intelligence method provides an anomaly detection strategy that captures the temporal and spatial link between several battery characteristics to increase battery life and system safety (*Das et al., 2021*). Any divergence from the nominal torque signal corresponding to the current task (defined by a specific set of waypoints—position signals) would indicate an anomaly in the robot's performance (*Aivaliotis et al., 2023*). Additionally, IoT devices continuously gather energy and meteorological data, which the model carefully processes and accurately examines in real-time to find any potential irregularities in energy consumption (*Malki, Atlam & Gad, 2022*).

### Remaining useful life estimation

The RUL is the time frame from the present until the expiration of the useful life (*Cai et al., 2022*; *Han, Li & Chen, 2023*; *Hu et al., 2023b*; *Mitici et al., 2023*; *Shaheen, Kocsis & Németh, 2023*; *Wang et al., 2023*; *Yang et al., 2023*; *Yousuf, Khan & Khursheed, 2022*; *Zhu et al., 2023*). RUL has been used in numerous industries, such as rotating equipment, batteries, and aerospace, to name a few, to warn operators of early failures (*Zhao et al., 2020b*). According to a preliminary study conducted by *Kang, Catal & Tekinerdogan (2021)*, the RUL is primarily utilised as a risk indication. It suggests how long a machine may be used properly before breaking down. Run-to-failure data from machine operations must be collected for RUL modelling. However, this task is challenging. Knowledge-based models (KBM), PM, DDM, and DL are possible areas for study on RUL prediction. In KBM, experts define the rule sets and evaluate the equipment's state based on historical failures, occasionally leading to disparities (*Garga et al., 2001*).

## Implementation of maintenance management

According to the preliminary studies by *Lopes et al. (2016)*, the term "maintenance management" refers to a group of tasks that specify maintenance goals, plans, and roles as well as how they are conducted. These tasks include planning, controlling, and monitoring maintenance, enhancing organisational procedures, and considering economic factors. To increase productivity and lessen the effects of unplanned downtime, maintenance management uses a variety of methods and strategies, such as failure mode and effects analysis (FMEA), total productive maintenance (TPM), and reliability-centred maintenance (RCM) (*Oliveira, Lopes & Figueiredo, 2012*). Similarly, *Fraser, Hvolby &*

*Tseng (2015)* discovered that the most widely used management models are condition-based maintenance (CBM), RCM, and TPM. TPM aims to achieve minimal malfunctions, accidents, and faults in the production system by improving the efficacy of the equipment (*Alkhoraif, Rashid & McLaughlin, 2019*). Companies can improve their competitive position by offering their consumers lower pricing and faster delivery times when implementing TPM programmes successfully (*Teera-achariyakul & Rerkpreedapong, 2022*). RCM is a technique for maintenance that emphasises system dependability (*Keynia et al., 2022*). To create proactive maintenance plans that increase equipment performance and dependability, RCM concentrates on comprehending failure modes and their effects. RCM has also been extensively applied in industry to lower maintenance costs (*Geisbush & Ariaratnam, 2023*). According to the preliminary studies by *Filz et al. (2021)*, a popular technique for identifying fault conditions in parts, components, or larger capital goods in advance is FMEA. With the goal of averting mistakes and failures during the development stage, FMEA is frequently employed in the product development and design phases. This can improve a system's overall reliability as well as the dependability of its individual parts. Based on the system status, which is determined by the rate of deterioration or level of performance, CBM chooses a maintenance action. Of all the maintenance types, CBM has drawn the most attention since it avoids needless maintenance actions, reducing costs while also considering safety (*Bousdekis et al., 2018*; *Shi et al., 2020*). Computerised Maintenance Management System (CMMS) to facilitate increased performance and dependability. It is an efficient computerised process management system (*Wienker, Henderson & Volkerts, 2016*). A CMMS is a standardised platform combining information from TPM, RCM, FMEA, CBM, and procedures. Other than that, it guarantees that maintenance activities are in accordance with the organisation's goals and priorities, facilitates data-driven decision-making, and expedites maintenance operations.

As provided in Table 4, the research presents that five studies apply CBM to maintenance management methods and strategies (*Avornu et al., 2022*; *Centomo, Dall'Ora & Fummi, 2020*; *Harries et al., 2023*; *Mrzyk et al., 2023*; *Mubarak et al., 2022*). Three studies deal with the application of FMEA (*Aivaliotis, Georgoulias & Chryssolouris, 2019*; *Mourtzis, Tsoubou & Angelopoulos, 2023*; *Werner, Zimmermann & Lentes, 2019*), two studies with CMMS (*Hosamo et al., 2022*, *2023*), and only one study with the application of RCM methods and strategies (*Mubarak et al., 2022*). However, no study uses TPM techniques and strategies.

Typical difficulties and obstacles to putting maintenance management into practice are unplanned maintenance, occurred when an unforeseen breakdown happens, which is bad for the organisation. The management always feels that spending on labour, inventory, and service costs is superfluous. Hence, these expenses always run counter to the budget. It is vital to shift from reactive maintenance to PdM to accomplish this. Effective time management is crucial for producing high-calibre and effective work. Two cutting-edge technologies that could completely change how equipment and system maintenance are managed are PdM and DT. AI techniques like ML and DL can further the research of equipment or system performance forecasting if sufficient data from maintenance histories exists to build the most accurate model feasible. Therefore, PdM and DT models must be

proposed in all industries to further increase the availability and reliability of the equipment and maximise the efficiency of the organisation. Accurately determining the actual times of issue occurrence, reporting, and resolution is made easier with real-time monitoring. Organisations may effectively handle reoccurring difficulties and become more proactive with their response strategies by figuring out these times. There are numerous advantages to implementing maintenance management practices. It reduces operational disruptions and costly maintenance by preventing equipment breakdowns and unforeseen downtime. Other than that, it promotes worker and user safety, optimises the equipment's performance, and prolongs its lifespan. This implementation of maintenance management gives the PdM study using DT significance for application in various industries.

## DISCUSSION

### Outcomes and research gaps/future work for PdM using DT

The current literature on PdM utilizing DT is succinctly synthesised in Table 4, outlining findings and highlighting avenues for further investigation. The collective body of research underscores the affirmative outcomes of employing PdM through DT. Each study introduces distinct DT models and techniques, contributing to the diversity and richness of approaches within this domain. Moreover, identifying research gaps and prospects for future exploration varies across studies, contingent upon specific recommendations tailored to diverse sectors. Despite the variations in the implementations of DT and PdM and the existence of research gaps and areas for future study, the result suggests that this idea has been effectively implemented in three primary domains. These concerns were also essential in managing intricate equipment or systems within the primary three domains of facilities.

PdM and DT are two emerging technologies that possess the capacity to fundamentally transform equipment or system administration maintenance. AI-assisted algorithms for predicting the performance of systems or equipment can be developed if sufficient maintenance history data is available to construct the most precise model. Moreover, numerous industries can benefit from a predictive analytics model that predicts potential defects, increases equipment utilisation, and reduces future failures. Alternatively, DT can provide a virtual representation of a physical system or piece of apparatus or system so that its performance can be monitored and its maintenance requirements can be anticipated. It enables real-time monitoring and analysis of equipment or system behaviour, allowing maintenance teams to identify issues before they become critical.

Integrating DT and PdM could promote the transition of equipment or system maintenance teams from a reactive to a proactive maintenance strategy. This transition leads to enhanced equipment or system uptime and patient safety while simultaneously reducing costs. These technologies allow maintenance teams to identify possible issues in a timely manner, enhancing the overall dependability of equipment or systems and decreasing the probability of failure during operation. This article provides a novel insight into combining data-driven PdM with DT modelling to address maintenance management in healthcare, utilities (smart water management), and agriculture (smart farm).

## Elevating industries with integrated PdM and DT solutions
### PdM using DT for medical equipment maintenance management

In the ever-evolving realm of contemporary healthcare, the proficient administration of medical equipment maintenance emerges as a pivotal element in guaranteeing the highest standard of patient care and operational effectiveness. Integrating PdM and DT technologies offers a paradigm-shifting approach to improve maintenance practices for medical equipment. Moreover, the integration of PdM and DT allows for proactive problem prediction, enhances the effectiveness of troubleshooting, optimises maintenance scheduling, and enables decision-making based on data analysis. Drawing upon the findings obtained from the systematic evaluation of relevant literature, a fully integrated framework is proposed, which combines the capabilities of PdM and DT technologies, as illustrated in Fig. 9. This framework functions as a crucial blueprint that coordinates the smooth integration of diverse components to enable effective management of medical equipment maintenance in healthcare environments. Furthermore, the core of the integrated framework is comprised of a precisely crafted system architecture. The architectural design functions as a central point of connection, facilitating the integration of many components essential to the maintenance management process. The system is continuously supplied with a flow of data streams, each contributing to a comprehensive comprehension of the status and operation of the medical equipment. However, the most essential data source for this integrated system is the physical medical equipment.

The aforementioned entity serves as the fundamental component of the maintenance ecosystem, playing a crucial role in supporting the overall structure. In addition to the physical equipment, there are data streams created by sensors. The real-time data inputs are obtained *via* a network of embedded sensors that are strategically placed on the medical equipment. These inputs provide a live view of the operating condition of the equipment. The framework further incorporates a DT model, serving as a virtual depiction of the medical equipment. The digital replica has diverse attributes, qualities, and actions that closely resemble those of its tangible counterpart. The virtual model functions as a dynamic representation that has the ability to simulate real-world circumstances and forecast future outcomes. Data technology plays a crucial role in enabling the use of predictive analytics and real-time monitoring. In addition, DL algorithms can be used to represent DT or predictive analysis, including Health Index (HI), predictive model, and fault diagnosis or prediction of RUL.

The architecture for PdM of medical equipment, known as the DT architecture, is constructed based on actual physical medical equipment and their corresponding operating environment. Various kinds of sensors are often installed on physical medical equipment. These sensors are crucial in providing data for data-driven operations such as RUL prediction and component failure detection. Sensors are used to gather real-time operating data for medical equipment, which is then employed to develop a DT model or virtual model of the item using simulation data. Furthermore, the use of sensor technologies for the collection and transmission of data from medical equipment to a central system enables the instantaneous remote monitoring of the operational data of

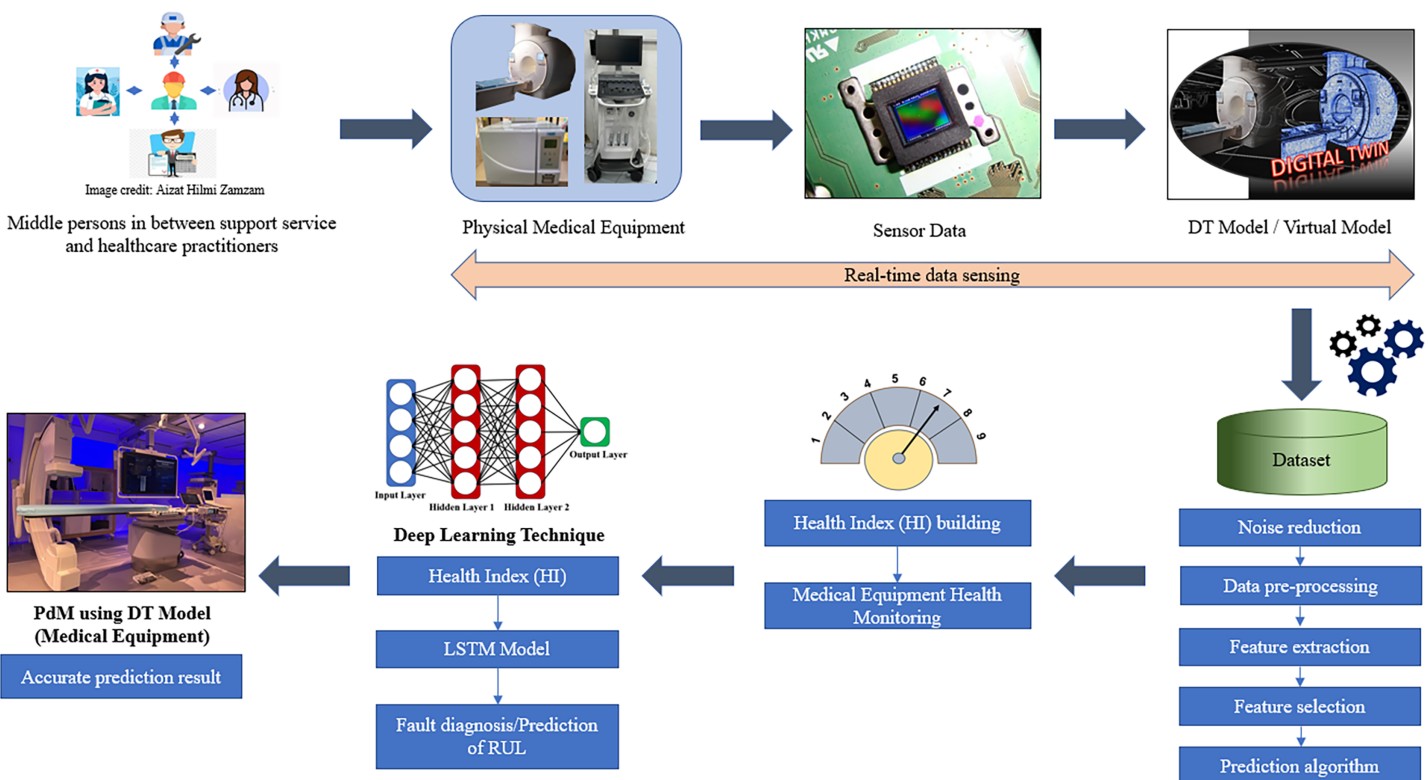

**Figure 9** **The proposed framework of DT using the PdM framework in the healthcare industry.** Image source credit: middle persons in between support service and healthcare practitioners, Aizat Hilmi Zamzam. Photo credit (Physical Medical Equipment images, DT Model/Virtual Model, PdM using DT Model): Nur Haninie Abd Wahab.               

such equipment. This technology facilitates the remote monitoring of medical equipment by healthcare practitioners, allowing for tracking usage trends and timely detection of possible issues or failures. Through the real-time monitoring of operational data, healthcare practitioners have the ability to proactively detect and address issues before they escalate, reducing the probability of medical equipment failure or outage.

The model is founded upon empirical facts, with the functioning of medical equipment yielding a substantial volume of data. The use of a data-driven approach mandates that prior sensor data undergo a series of procedures prior to its application in prediction tasks. These procedures include noise reduction, data pre-processing, feature extraction, and state detection. Creating hybrid simulation models that strike a compromise between computational economy and accuracy by fusing data-driven and physics-based modelling. These are some methods for coming up with creative techniques to maximise the computational effectiveness of real-time DT simulations. By processing data closer to the origination point, edge computing lowers the latency in transmitting data to a central server. Lower latency allows for faster analysis of sensor data and quicker reaction to possible maintenance issues, which is crucial in PdM, where rapid insights are required.

The integrated system effectively collects real-time data from physical equipment and sensors, including performance measurements, operational parameters, and environmental factors. The data stream is further processed using sophisticated analytical techniques to derive a HI. It serves as a measurable indicator of the equipment's comprehensive health and operational soundness (*Yin et al., 2020*). Note that the state of medical equipment is tracked using a HI for PdM to determine when maintenance is required. A HI is usually created by analysing data obtained from sensors and other monitoring equipment that record numerous factors. First, the equipment's health factors were identified, and its state of health was assessed. The HI-based prediction model uses the input data and labels it generates (*Zheng et al., 2021*). DL methods are employed, including ANN, LSTM, and DNN, amongst others (*Neo et al., 2022*). DL models can be trained to forecast the need for maintenance, enabling maintenance personnel to plan replacements or repairs before a significant breakdown occurs. The DT model/virtual model, driven by DT for PdM of medical equipment, has assessment, optimisation, prediction, and other capabilities, such as fault detection and RUL prediction of medical equipment, as well as a conceptual model. Organisations can facilitate the efficient transfer of knowledge for applying PdM using DT in the medical industry through various strategies. For example, developing comprehensive training programmes for employees at all levels, including engineers, data scientists, medical professionals, and maintenance staff. Other than that, providing practical training that covers the principles of PdM, the development and use of DT, and the specific requirements and intricacies of the medical industry. Personnel involved in using PdM insights using DT for decision-making in the medical industry should have a combination of technical, domain-specific, and operational skills. Personnel must comprehensively understand the medical devices and systems to be monitored. This includes knowing how the devices work, their critical parameters, and potential failure modes. Training in collaboration and communication between different disciplines is essential. Personnel should be able to effectively communicate findings and recommendations to medical professionals, engineers, and decision-makers.

ML and DL were utilised, each showcasing unique advantages. A comparative analysis was performed, revealing ML's superiority over DL in accuracy, precision, specificity, and F1 score. Although DL has a shorter training time, its accuracy is lower than that of ML (*Rahman et al., 2023*). However, according to the preliminary studies by *Zhai & Qiao (2020)*, DL's excellent performance is primarily due to a massive amount of training data and a deep network topology. The proposed framework employs DT technology to detect and diagnose problems and predict medical equipment components, enabling maintenance personnel to make more informed decisions at the appropriate moment. This will enhance the models to create the ideal strategies for future medical equipment in the healthcare industry. Several technical obstacles and challenges can arise when seamlessly integrating these technologies, such as medical facilities being subject to constant changes such as equipment upgrades and replacements and workflow changes.

To ensure that DT remains accurate and current in a dynamic medical environment, a robust mechanism for continuous updating and adaptation is required. In addition, collaboration between healthcare providers, vendors, medical device manufacturers, and

regulatory bodies is essential for successfully implementing PdM and DT. Coordinating the efforts and expectations of these different stakeholders can be challenging due to differing priorities, timelines, and business models.

The significant outputs of the DT model will maximise operating costs within budgeted costs by improving medical equipment maintenance operations through the implementation of effective maintenance management practices. Moreover, accidents, financial losses, and fatalities can be avoided by scheduling maintenance in advance when it is acknowledged that medical equipment will fail. Medical equipment significantly contributes to the effectiveness of healthcare quality (*Zamzam et al., 2021a*). In addition, the primary goals of maintenance are dependability, maintainability, availability, and safety. Hence, medical equipment should not fail frequently and should be repaired as soon as a flaw is discovered (*Abd Rahman et al., 2023*).

This study has some limitations despite producing good performance prediction results. Most studies only evaluate monitoring data without fully examining the equipment's real-time state, leading to biased results and low real-time forecast accuracy. As a result, the relationship between the actual operational state and the virtual simulation must be actualised using real-time monitoring data. It will aid in comprehensively forecasting equipment health conditions (*Yang et al., 2021*). Thus, the proposed framework can be embedded with the existing national standard of handling medical equipment.

For example, the concept of data-driven DT-PdM can be adopted in any healthcare institution, such as an imaging department. DT can simulate the behaviour of imaging equipment, including Magnetic Resonance Imaging (MRI) machines, Computed Tomography (CT) scanners, and ultrasound machines. Analysing real-time data from these medical types of equipment makes it possible to develop PdM models that optimise maintenance schedules and prevent unscheduled downtime.

### PdM using DT for smart water management

As medical equipment health indices paved the way for timely interventions, intelligent maintenance strategies can monitor and optimise the urban equivalent, from water and energy distribution to transportation and waste management. Due to urbanisation, the world has recently confronted numerous issues, including urban poverty, exorbitant prices, traffic congestion, lack of shelter, absence of financial backing, increased crime, environmental degradation, and inequality (*O'Brien, Pike & Tomaney, 2019*). To meet the predicted growth in urbanisation, it is becoming increasingly difficult to design, build, run, and maintain urban infrastructure systems such as water distribution networks (WDN) with sufficient sustainability and resilience (*Wu et al., 2023*). The adoption of a Smart Water Grid (SWG) is an effective technique for ensuring adequate WDN sustainability and resilience (*Public Utilities Board Singapore, 2016*).

The proposed combined PdM and DT framework consists of a water distribution system (WDS), sensors, data collection and management from sensors and PdM using the DT model (see Fig. 10). The purpose is to track drinking water quality and pipeline leaks and exchange this data in order to save water and enhance the present WDS. The DT architecture for PdM is built on real physical WDS for pipe monitoring or the PipeSense

**Pipe Monitoring/PipeSense System**

**Figure 10 The proposed framework of DT using the PdM framework in the smart water management.**

system of a physical layer. In physical WDS, numerous types of sensors have been placed in water treatment facilities as well as around the pipe monitoring or PipeSense system, and these sensors then provide information for data-driven analysis. The sensors displayed in Fig. 10 are water quality sensors and pH and flow sensors that serve as measurement objects of the WDS layer. The idea of numerous networked sensors on equipment, individuals, and products, along with intelligent monitors, is becoming more attainable due to the advancements in communication technology (*Olsen & Tomlin, 2020*).

The sensor data layer of this structure includes sensor data collecting and management to analyse the acquired data. This involves network data transfer as well as server storage and database administration. Sensors collect operational data in real-time for WDS, and simulation data is used to generate a PdM using the DT model. Furthermore, implementing real-time remote monitoring will allow for more efficient decision-making for PdM using DT. These technologies could serve as the basis for the PdM using DT of the city and help in the sustainable design of smart cities. With this technology, facility management staff or users can remotely monitor water system operation, examine usage patterns, and spot potential faults or malfunctions in real-time. The model is based on

data, and the operation of the water network generates a large amount of data. In the data-driven method, the previous sensor data must go through a series of steps before it can be used for prediction, including cloud data storage, communication, data analysis, and processing.

Businesses can use a range of approaches to facilitate knowledge transfer that will guarantee the effective use of DT for PdM in smart water management. Other than that, encourage the sharing of knowledge through unofficial channels so team members with less experience can benefit from their mentors' real-world expertise. In order to properly analyse insights from PdM and DT for smart water management decision-making, staff members need to be trained in problem-solving and critical thinking. Hence, they will gain communication skills to communicate technical findings effectively and clearly to stakeholders who are not technical.

Optimising the computational efficiency of real-time simulations with DT in large-scale and complex industrial environments, especially in smart water management, requires the consideration of several innovative approaches and frameworks. Therefore, integrating ML algorithms aid in predicting system behaviour and optimise simulation parameters. Train models using historical data to make predictions, facilitating real-time adjustment of simulation parameters. This enhances computational efficiency for improved performance. With edge computing and distributed processing, real-time analytics for PdM is made possible in large, geographically dispersed businesses like smart water management. By processing data closer to the source, edge computing lowers the delay in sending data to a central server. In smart water management, where prompt problem-solving and prevention are paramount, reduced latency guarantees prompt sensor data processing for PdM.

The application level is the highest level in the water management system's framework. Data from sensors can be enhanced by models to create useful PdM using DT for better maintenance decision-making. To produce a more precise forecast for pipe monitoring or the PipeSense system, the anticipated circumstances are then updated using the observational data collected by the system to produce an accurate prediction result.

An effective strategy to ensure PdM using DT according to the standard of the proposed approach can be incorporated into the current national standard for maintaining water treatment assets for Smart Cities System maintenance. Therefore, urban drainage terminology and actions to establish a DT through constructing digital ecosystems and open data standards format are crucial in the water sector (*Pedersen et al., 2021*). The integration of DT for PdM in smart water management is promising. Nevertheless, companies may encounter unforeseen technological hurdles and difficulties. These difficulties include accurate prediction depending on the availability and quality of real-time data from sensors and other devices. This data can sometimes be delayed, erroneous, or incomplete. Hence, the protection of confidential information is an important issue, especially when it comes to vital infrastructure such as water management. When using DT, users can transmit sensitive information. Furthermore, several rules and regulations apply to the smart water business, and adhering to them might be difficult. Establishing unambiguous communication channels with regulatory bodies, keeping up a continuous

dialogue to comprehend and fulfil compliance obligations, and actively engaging in industry standardisation initiatives are all crucial. The successful deployment of PdM and DT frequently necessitates cooperation between several entities, such as regulators, technology vendors, and water utilities. Encourage a culture of cooperation, form alliances and consortia, and take part in industry coalitions state expectations for cooperation as well as roles and responsibilities.

A limitation of this study is that a large data set must be collected, which requires pre-processing to determine if the data can be used. Data pre-processing involves checking the data for errors such as missing or duplicate records. In addition, DT necessitates collecting enormous amounts of data from various endpoints, each providing a possible point of vulnerability (*Wu et al., 2023*).

A practical and affordable alternative, an integrated sensor platform, is offered by PipeSense. Monitoring the WDS and locating leakage through real-time monitoring data or DT is possible. The idea of data-driven DT using PdM, for instance, can be used in any system for smart cities, particularly in the field of WDS, for example, real-time monitoring of asset conditions. This proposed model contributes to innovative maintenance management using the concept of PdM with DT. The advantages that DT offers in implementing PdM in the water industry in terms of digital transformation and improving system performance are substantial and immediate impacts on customer satisfaction in water services, cost efficiency in asset maintenance, and ensuring greater upkeep and environmental safety. In addition, DT enables estimation of utilisation during periods of high demand, improves water flow and pressure control, and significantly increases system dependability and flexibility during operation (*Ramos et al., 2022*).

### PdM using DT for smart farm maintenance management

Population growth has caused a sharp increase in food consumption, and agricultural mechanisation has become a key strategy for enhancing grain yield and quality (*Mantoam et al., 2020*). The machinery of agriculture is becoming a fundamental and important equipment in modern farming (*Han et al., 2020*). Regular maintenance is required to avoid losses caused by failures, and firms deploy maintenance vehicles to assure stability in agricultural machinery operations (*Wang, Hu & Ren, 2021*). Modern agriculture is not possible without accurate and current information regarding the farming process. Nevertheless, farms are being forced to concentrate a growing amount on digital technology, like measuring and monitoring tools, sophisticated data analysis, and smart machinery (*Verdouw et al., 2021*). As a result of the quick growth of technologies like computing in the cloud, big data, the IoT, ML, virtual reality, and the field of robotics, agricultural production is quickly shifting towards smart farming technologies (*Kamilaris & Prenafeta-Boldú, 2018*; *Zhai et al., 2020*).

The proposed architecture of PdM with DT for smart farm equipment is divided into three layers: the physical, communication protocol, and application layers. Each layer is important for obtaining a good model with accurate results (see Fig. 11).

At the physical level, various sensors are installed in the smart agriculture field, where data is collected using sensor applications. The proposed framework has two types of

**Smart Farm Equipment**

**Figure 11 The proposed framework of DT using the PdM framework in smart farm equipment.** Photo credit: Nur Haninie Abd Wahab.     

sensors: environmental and camera. Camera sensors work to detect the farm condition, while environmental sensors are smart farm characteristics such as temperature, humidity, air velocity, light, and ventilation (*Sung & Kim, 2022*). Multiple types of sensors are strategically implemented into agricultural machinery, generating a vast amount of real-time operating data that serves as the foundation for data-driven analysis (*Javaid, Haleem & Suman, 2023*). The sensors are used to generate real-time data for smart farming and store it in the database for the next shift process. Furthermore, real-time remote monitoring will enable more efficient decision-making for PdM using DT. These developments could serve as the basis for PdM using DT of farm equipment and help develop sustainable farm equipment. Hence, users or farmers can use this technology to monitor farm equipment remotely, investigate consumption trends, and detect potential defects or malfunctions in real-time.

Optimising the computational efficiency of real-time DT simulations in large-scale and complex industrial environments, especially in the smart farm industry, requires the consideration of innovative approaches and frameworks. Thus, real-time data compression techniques are implemented to reduce the amount of data transferred between sensors and the central system. The use of intelligent filters to transfer only the information is essential for running simulations and reduces the computing load. In sectors like smart farms with extensive and dispersed infrastructures, edge computing and distributed processing are essential to providing real-time analytics for PdM. It keeps

private data inside the local infrastructure, which helps alleviate privacy worries. Consequently, encryption and secure communication protocols guarantee the confidentiality and integrity of information shared among dispersed nodes.

Next is the communication protocol layer, where data is transferred to the communication protocol to analyse and visualise the pre-process. In this process, observation of missing data and data cleaning is conducted. In addition, normalisation of the data was necessary to guarantee that the gap between every characteristic value was equally weighted (*Zamzam et al., 2021b*). This preliminary process will also send this data to the data analytics layer. A PdM using the DT model is created at the application level from the analysed, integrated, and visualised data for improved maintenance decisions. The model can provide accurate outcome predictions for smart farm equipment monitoring. In addition, the authors propose to call this model the model for intelligence, efficiency, and visibility.

The contribution to the creation of this model in the agricultural field is that it can help the farmer or user increase crop yield by increasing equipment uptime. PdM using DT can help predict equipment damage that occurs and reduce maintenance costs. Organisations that adopt the DT can gain major advantages such as improved processes, reduced time to the marketplace, and innovation in goods and services (*Ante, 2021*; *Javaid et al., 2022*; *Vitorino et al., 2019*). Businesses can guarantee that staff members possess the skills and knowledge required to apply PdM using DT in the smart farm sector by employing various approaches, including cross-disciplinary workshops, external training opportunities, internal workshops and seminars, and collaborative learning platforms. Maintaining the workforce's current knowledge of cutting-edge technology and industry best practices requires ongoing training and upskilling. Therefore, workers in smart farms must possess crucial knowledge and experience requirements, including familiarity with agricultural operations, maintenance procedures, and the ability to convert PdM knowledge into actionable decisions.

The usage of sensors in individual equipment to gather data has helped IoT applications advance. Nevertheless, discrete sensor and internet covering of every location resulted in substantial facility management expenses. Moreover, adherence, safety, and trustworthiness standards are becoming more stringent and subject to rapid change (*Masmoudi et al., 2016*).

Using applications for real-time monitoring equipment, the model can be integrated with PdM, utilising DT to facilitate decision-making. The user and farmer can comprehend the data without difficulty. The concept of DT in smart agriculture is still in its earliest stages. At the beginning demonstrating stage, numerous farmers are exploring incorporating smart technology and methods that enhance the effectiveness of agricultural processes (*Verdouw et al., 2021*). Agricultural mechanisation can save labour time and other resources, enhance labour efficiency, and lower production costs (*Mu et al., 2018*). In addition, the proposed technique can be implemented in the existing national standard for agricultural machinery maintenance as an efficient method to ensure PdM with DT in accordance with the standard. DT integration of PdM in the smart agricultural sector could encounter several technological challenges. Variations in the weather, plant health,

and pest infestations are only a few examples of the dynamic and unpredictable variations that affect agriculture the creation of adaptable DT models that can be modified on the fly. Hence, it establishes feedback and real-time monitoring systems to inform the DT about the changing agricultural landscape. In addition, effective implementation of PdM and DT often requires collaboration between different organisations, including farmers, technology providers, and regulators. These organisations can foster a culture of collaboration, establish partnerships, and participate in industry consortia or alliances. Clearly define roles, responsibilities, and expectations for collaboration.

Five review articles are included in this discussion and discuss PdM using DT. There are different aims for all the papers. The first review article aims to redefine the concept of next-generation-Digital-Twin (nexDT). Furthermore, it introduces a specialised term for electrical machines manufacturing, PdM, and control, summarising the majority of pertinent work in the process. Other than that, it offers a new definition unique to PdM and serves as a basis for future efforts (*Falekas & Karlis, 2021*). This study reviews the most recent descriptions found in the broader literature and addresses the open challenge given by the indirect usage of an established Digital Twin Framework (DTF) in industry, specifically system security and risk assessment. Aspects such as goals, application areas, platforms, representation types, approaches, abstraction levels, patterns of design, protocols for communication, twin parameters, challenges, and solutions are highlighted in the second review article on PdM through the use of DT (*van Dinter, Tekinerdogan & Catal, 2022*). There are 42 primary studies that have been evaluated. The study revealed that the computational effort, the variety of data, and the complexity of the models, systems, or components represent the greatest challenges in the development of these models. Consequently, a presentation is given on using a DT framework in PdM and its expansions *via* physics-based modelling and ML. Planning for repair, logistics for deployment, and performance assessment measures are all included in fleet management (*Kunzer, Berges & Dubrawski, 2022*). This study attempts to clarify the definition of the DT by looking at the term's history and original context in asset maintenance and fleet management, planning, operations and product lifecycle management. The challenges for this study are safety protocols, the adoption of the DT framework in the workplace, the robustness of the sensors, missing data, poor quality data, and offline sensors. Review four examines the modelling techniques, the application framework, and the relationship between the virtual image and the physical object as recent developments (*You et al., 2022*). This study aims to describe the benefits of the PdM using the DT paradigm that are thought to exist and examine the methods and applications for each category. A total of 30 primary studies were examined. Four critical challenges for DT-driven PdM discussed in this study are the standardisation of the framework, holistic evaluation methods, the need for a digital model with high fidelity, and a multi-level model and multi-component. Note that the creation and assessment of reference architectures created with the use of well-known software architecture techniques are the topics of the final review article (*van Dinter, Tekinerdogan & Catal, 2023*). The study revealed three viewpoints for DT-based PdM systems. The authors developed a context diagram for the user view, and for the structural view, they created a package diagram. The authors also created an application

architecture for each case study based on the features of the study using each reference architecture view, a layered view to illustrate the system's breakdown into layers, and a deployment view to depict the hardware, software, and surrounding environment. There are 42 selected primary studies for which a trait model was developed. This study is about the continuous consideration of challenges in production, such as changes in data distribution, skewing of training data, or problems with data quality.

From all the highlighted review articles, this study differs by focusing on three areas of PdM-based DT implementation, namely healthcare, utilities (smart water management), and agriculture (smart farm), and fulfilling the objectives of this study. This work analysed 34 articles from 2018 to 2023 and discussed various challenging PdM with DT. In addition, an explanation of the use of sensor types for the relevant industries, and finally, the implementation of maintenance management, which focuses on the diversity of maintenance management methods and strategies, issues proposed in the study, PdM and DT methods, studies conducted in real-time monitoring, and the importance of research.

The integration of DT and PdM technologies has resulted in industry-altering advancements. This integration enables proactive equipment maintenance in healthcare management by simulating real-time conditions and anticipating potential malfunctions. DT and PdM optimise the operation of infrastructure in utilities (smart water management) by integrating real-time sensor data with predictive analytics to anticipate leakage and system inefficiencies. Similarly, in agriculture (smart farm), the combination of DT and PdM aids in the surveillance of apparatus in real-time, while predictive insights guide opportune productivity-enhancing interventions. These applications demonstrate the value of DT and PdM in revolutionising maintenance practises across industries, thereby paving the way for enhanced operational efficiency, resource optimisation, and overall performance.

However, implementing PdM and DT to maintain equipment or systems can present various challenges. Here are some of the major considerations to address prior to adopting the concept:

1) **Data collection and management:** Collecting and managing the enormous quantities of data required for PdM and DT can be difficult, particularly for aged equipment or systems not necessarily designed to collect and transmit data. Facilities must invest in automated data acquisition and management tools to ensure accurate, trustworthy, and secure data. To achieve a balance between optimising network maintenance through PdM and DT and meeting stringent regulatory requirements, meticulous planning and implementation are required.

2) **Data integration:** It can be difficult to integrate data from multiple sources, such as equipment sensors, electronic medical records, and patient feedback. A data integration strategy must be developed by facilities to ensure that all data sources are integrated and accessible for analysis.

3) **Model development:** Developing accurate PdM models and DT requires an in-depth comprehension of the behaviour and performance of equipment or systems. To develop and evaluate these models, facilities must invest in expertise and resources.

4) **Compliance with regulations:** PdM and DT technologies must comply with the country's own regulations, which can present additional challenges for facilities. It is essential to ensure that these technologies adhere to regulatory requirements and that data is kept secure. In healthcare, utilities (smart water management), and agriculture (smart farm), the need to adhere to sector-specific rules can confound the integration of PdM and DT. Utilising these technologies in healthcare, patient data privacy statutes such as the Health Insurance Portability and Accountability Act (HIPAA) and General Data Protection Regulation (GDPR) must be adhered to. Similarly, water networks must comply with water quality regulations and data security. In agriculture, machinery optimisation must consider environmental regulations and data privacy. It is essential for the safe and effective implementation of PdM and DT methods in each field that they adhere to these regulations.

5) **Cost:** Implementing PdM and DT technologies can be costly, necessitating substantial investments in hardware, software, and personnel. The costs and benefits of these technologies must be thoroughly evaluated by facilities to determine if they are worth the investment.

6) **Changes to organisational cultures:** Adopting new technologies and modifying maintenance practises can be difficult for organisations, necessitating a shift in organisational culture and perspective. Facilities must invest in change management and training to ensure their employees are prepared to implement these technologies and procedures.

Across all industries, integrating PdM and DT technologies presents obstacles. This involves managing immense quantities of data, integrating diverse data sources, creating accurate models, and ensuring compliance with sector-specific regulations such as HIPAA, GDPR, water quality standards, and agricultural regulations. Implementing PdM and DT can be expensive, necessitating a thorough cost-benefit analysis and a cultural transformation in organisations, necessitating change management and training for successful adoption. Therefore, implementing PdM and a DT to maintain three potential applications for equipment or systems requires meticulous planning, investment, and specialised knowledge. Facilities must be prepared to surmount these obstacles To reap the benefits of these technologies, which include improved equipment or system uptime, reduced costs, and increased patient or worker safety.

## CONCLUSIONS

Early maintenance planning significantly reduces the probability of failures, financial losses, and even fatalities by enabling more accurate prediction of delays. PdM is vital for organisations, delivering cost-effective maintenance, extended equipment lifespan, and improved safety. Hence, integrating it with DT offers immense potential. However, its application in crucial sectors like healthcare, utilities (smart water management), and agriculture (smart farm) remains unexplored. Although these industries have the potential to benefit greatly from the integration of PdM and DT, there is a dearth of comprehensive research in this area. This article analysed 34 articles from 2018 to 2023, identifying gaps in

PdM using DT. Four major themes emerged from the research: various model categories, diverse approaches, predictive outcomes, and implementation of maintenance management. It is discovered that integrating supervised learning with diverse algorithms has been identified as the most efficacious approach to assessing equipment performance, providing practical insights that can inform maintenance decisions. DT-driven models improve maintenance decisions and help optimize operational costs within budgets. They facilitate the implementation of cost-effective equipment maintenance and enhance overall reliability by offering real-time data insights. Nevertheless, the absence of studies in healthcare, utilities (smart water management) sectors, and agriculture (smart farm) signifies a critical gap. Therefore, recommendations for implementing PdM using DT in these domains are crucial for progress. Concrete suggestions on leveraging DT to enhance systems in these sectors offer a pathway for practical solutions based on real-time data. To address this research gap, suggesting PdM utilising DT in these sectors is necessary. These initiatives hold the potential to revolutionise maintenance practices, enhance system optimisation, and provide effective solutions. The study's findings are of immense value as they indicate the possibility of groundbreaking progress, especially in unexplored industries. Although PdM with DT exhibits significant potential, its application in critical industries such as healthcare, utilities (smart water management), and agriculture (smart farm) remains largely unexplored. However, there is a limitation in implementing PdM with DT, where every equipment or system must be monitored in real time with sensors. Therefore, only facilities that have internet coverage can implement this PdM with DT. To overcome these limitations, facility management will incur significant costs to implement PdM with DT comprehensively. Future research that addresses this knowledge deficit will facilitate the development of practical solutions, such as improved equipment dependability and optimised maintenance procedures, which will have substantial societal benefits.

### Funding
The authors received support from the Malaysian Ministry of Health and the Hadiah Latihan Persekutuan (HLP) Scholarship was awarded to Nur Haninie Abd Wahab. The funders had no role in study design, data collection and analysis, decision to publish, or preparation of the manuscript.

### Grant Disclosures
The following grant information was disclosed by the authors:
Malaysian Ministry of Health and the Hadiah Latihan Persekutuan (HLP) Scholarship.

### Competing Interests
Kai Huang is an employee of the JiangSu XCMG HanYun Technologies Co., LTD., Xuzhou, China.

## Author Contributions

- Nur Haninie Abd Wahab conceived and designed the experiments, performed the experiments, analyzed the data, prepared figures and/or tables, authored or reviewed drafts of the article, and approved the final draft.
- Khairunnisa Hasikin conceived and designed the experiments, performed the experiments, analyzed the data, prepared figures and/or tables, authored or reviewed drafts of the article, and approved the final draft.
- Khin Wee Lai conceived and designed the experiments, analyzed the data, authored or reviewed drafts of the article, and approved the final draft.
- Kaijian Xia conceived and designed the experiments, analyzed the data, authored or reviewed drafts of the article, and approved the final draft.
- Lulu Bei performed the experiments, analyzed the data, authored or reviewed drafts of the article, and approved the final draft.
- Kai Huang performed the experiments, analyzed the data, authored or reviewed drafts of the article, and approved the final draft.
- Xiang Wu performed the experiments, analyzed the data, authored or reviewed drafts of the article, and approved the final draft.

## Data Availability

This is a literature review.

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
