# Peer review of "Systematic review of predictive maintenance and digital twin technologies challenges, opportunities, and best practices"

_PeerJ Computer Science, doi:10.7717/peerj-cs.1943_

## Round 0.1 · original submission · Major Revisions

All authors see some merit in the paper, but raise some strong issues that need to be addressed in the paper revision.

**Language Note:** The review process has identified that the English language must be improved. PeerJ can provide language editing services - please contact us at copyediting@peerj.com for pricing (be sure to provide your manuscript number and title). Alternatively, you should make your own arrangements to improve the language quality and provide details in your response letter. – PeerJ Staff

·

Basic reporting

Abstract
The section is well written. The Gist of the background study is given precisely for Digital Twins (DT) and Predictive Maintenance (PdM) principles. The aim of the study is stated clearly. The statement of the methodology needs to be enhanced, like which tools are used for analysing the data and also mention the filters used for collecting the data.
Introduction
• The word “WE” should not be used in the research article; instead, the authors may use the study, the research, the authors, the researchers, etc.
• The background of the study can be enhanced in detail in terms of Digital Twins and PdM.
• Research gap needs to be enhanced by stating clearly the gap which leads to the objective,
• The research questions are explained well enough.
• Again, the tools used to analyse the data are missing. Add the details in the aim of the study. This will enhance the readability of the review article.

Experimental design

Survey Method
• While the article clearly mentioned the inclusion category, why are conference and review articles excluded?
• What are the filters that are used for filtering the data, such as which kind of search?
• Line 197 states, “two of four were excluded using other methods.” Which other methods?
• Eligibility and Quality Evaluation and Data Extraction are written well enough.

Validity of the findings

Results
• The outcome of the study is stated well enough, and I appreciate the good efforts in curating the figures. All the figures were curated in a manner that answers the research questions.
• All the models are explained clearly enough with a detailed explanation of the type of models used and their approaches.
• Predictive Outcomes were categorised clearly and explained in detail.
Discussion
• Findings are answered to all the research questions along with the previous study.
• The novelty of the study is stated clearly.
• I appreciate the efforts to state the implications of the study.

Additional comments

The findings of the results were clearly mentioned, along with the study's limitations. I appreciate the efforts in choosing the PdM, which is in need right now.
Overall, the article is clear enough but needs to justify the exclusion criteria and specify the tools for curating the figures and analysing the data. Filters used for collecting the data are not mentioned.

Reviewer 2 ·

Basic reporting

The authors aim to determine the significance of predictive maintenance utilizing digital twins by systematically reviewing prior research. The research offers new perspectives on predictive maintenance and digital twins, highlighting their emerging use in healthcare, smart cities and smart water management. The paper is relatively well organized, and it touches on a critical topic. The authors appear to address a clearly defined research question, conduct a comprehensive and systematic review of the literature, and use clearly reported, reproducible and systematic methods to identify, select and critically appraise relevant research. The survey and review methodology seem to be consistent with a comprehensive, unbiased coverage of the subject. Introduction section contain a well-developed and supported argument that meets the goals set out. The related sources are adequately cited and the review is organized logically into coherent paragraphs and subsections.

Some suggestions, questions and concerns about the paper are listed below:


1. The abstract misses to highlight “the need to write the review” although there are some survey and reviews (Predictive maintenance using digital twins: A systematic literature review; Digital Twin for maintenance: A literature review; Digital twin in electrical machine control and predictive maintenance: State-of-the-art and future prospects; Overview of predictive maintenance based on digital twin technology; Advances of digital twins for predictive maintenance; Digital Twins for predictive maintenance: A case study for a flexible IT-architecture, etc.) in the related subject. How this survey will contribute to the scientific body of knowledge?
2. What is the coverage (both temporal and domain) of the literature and how the literature was distributed across time/ domains?
3. Why are the specific filtering criteria used by the authors specially to finalize the set of literature for review?
4. Although major considerations and challenges are presented in “Discussion” section, “Conclusion” section does not identify unresolved questions / gaps / future directions.
5. The summary in “Approaches” section looks high-level. It would be nice to dig deep into more technical matters in this section.
6. Some figures (for example Figure 2 and Figure 5) should be polished.

Experimental design

As above

Validity of the findings

As above

Additional comments

As Above

Reviewer 3 ·

Basic reporting

The authors have given sufficient references to introduce the issues. However, I have not seen the explanation of other surveys about maintenance management in this paper. Does it mean there is no similar topic to that? If not, it would be better if the authors introduced the other surveys as well so that the contribution of this paper to maintenance management would be clear.

Experimental design

The study designed by the authors is tolerable to understand. The use of PRISMA in selecting the proper scholarly articles is also reasonable for this study. The authors are consistent in using the methodology

Validity of the findings

The lack of information about similar surveys in the maintenance management field makes the contribution of this paper not very clear. The authors should present other similar surveys to this study (if any). The authors have presented a gap for future works in the Conclusion section. However, the authors do not provide any recommendations for solving the gap. Elaboration of that recommendation is required for the readers.

Additional comments

Summary:
* * *
The authors propose a systematic review of predictive maintenance (PdM) and digital twin (DT) technologies challenges, opportunities, and best practices. The authors use PRISMA methodology in this study to select the related articles. The study reviews various modeling techniques to evaluate the lifetime of assets in terms of PdM. The authors use primary databases, namely Web of Science and Scopus, and additional databases, such as IEEE Explorer, Science Direct, Medline Complete, Emerald, Springer Link, and Dimension. The scholarly articles used in this study are the articles published from 2018 to 2023. The authors selected 14 scholarly articles in this study for the final selection. This study provides current and essential insights for academics, practitioners, and policymakers who want to utilize PdM and DT technologies in their respective domains successfully.

General comments:
* * *
The topic of this study is very interesting, especially for those who work in the maintenance management field. Some studies have been defined in this paper. So the readers can obtain some insights about the maintenance management field.

In terms of writing, the authors have tried to write this report well. However, some notes for this study can be considered:
1. Why did the authors decide the fields of healthcare, smart cities, and smart water management in this study? The motivation for choosing those fields should be elaborated
2. Why did the authors choose 2018-2023 as the range of search? How about the scholarly articles published before 2018? Were those available? The authors should give the reason for this in the manuscript
3. The number of selected articles (14 articles) is too few for a systematic review paper. Consider using other keywords to explore the more relevant scholarly articles
4. I think there is a typo, “encyclopaedias”, but I do not know what you mean
5. The paragraph “Hybrid Model” in line 315 should be placed on the next page
6. In the Conclusion section, the authors have not elaborated on the recommendations of PdM with DT in the three domains (healthcare, WDN, or agriculture). What are potential specific gaps in those domains, and what are the recommendations to address those gaps?
7. The use of color font in Figure 4 (the caption in the pie chart) should be changed to “white” (for example). So it will be clear and comfort to see
8. The caption of Y1 (Number of total publications from 8 databases (TP) and Y2 (Number of publications (fields)) in Figure 2 are not clearly explained. Please explain them
9. The word “Figure” is inconsistent between the main text and the attachment. For example, the line 154 uses “Fig. 1” to refer to “Figure 1” in the attachment

---

## Round 0.2 · accepted · Accept

The paper is ready for publication.

·

Basic reporting

The authors have improved the paper well as per the comments.
I appreciate their efforts.

Experimental design

The study design is good.

Validity of the findings

The authors have validated the findings significantly.

Additional comments

The paper still needs some improvements in the Discussion and Conclusion sections.
Discussion -- Please analyse, discuss, align and compare the present study findings with existing similar research.
Conclusion -- The authors may include the limitations of the study.

Reviewer 2 ·

Basic reporting

The authors did the required changes and enhanced the work as required.

Experimental design

The survey methodology seems to be consistent with a comprehensive, unbiased coverage of the predictive maintenance and digital twin technologies. To this reviewer, the authors have made satisfactory explanation and revisions according to the previous review comments.

Validity of the findings

The Conclusion now identifies unresolved questions / gaps / future directions. The authors did the required changes and enhanced the work as required.

Additional comments

Authors have explained well all issues and updated the manuscript properly.

Reviewer 3 ·

Basic reporting

The manuscript has improved than before. The basic reporting has been written well.

Experimental design

The manuscript has improved than before. The study design has been written well.

Validity of the findings

The manuscript has improved than before. The validity of findings has been written well.

Additional comments

In think there is a mispelling for "encyclopaedias" in the 1st paragraph of Identification section (line 203). Please correct that.

Annotated reviews are not available for download in order to protect the identity of reviewers who chose to remain anonymous.